# ELUCIDATING THE DESIGN SPACE OF LANGUAGE MODELS FOR IMAGE GENERATION

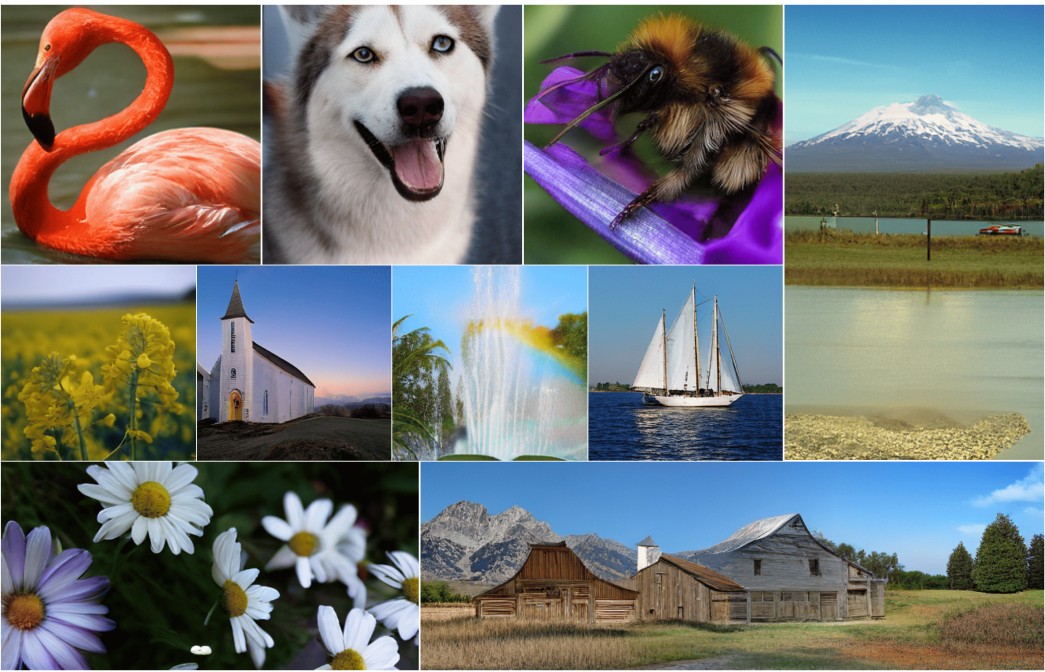

Figure 1: Generated samples from ELM-2B with 2-12 tokenizer trained on 256×256 ImageNet. ELM is flexible to generate *any-size* high-fidelity images.

## ABSTRACT

The success of autoregressive (AR) language models in text generation has inspired the computer vision community to adopt Large Language Models (LLMs) for image generation. However, considering the essential differences between text and image modalities, the design space of language models for image generation remains underexplored. We observe that image tokens exhibit greater randomness compared to text tokens, which presents challenges when training with token prediction. Nevertheless, AR models demonstrate their potential by effectively learning patterns even from a seemingly suboptimal optimization problem. Our analysis also reveals that while all models successfully grasp the importance of local information in image generation, smaller models struggle to capture the global context. In contrast, larger models showcase improved capabilities in this area, helping to explain the performance gains achieved when scaling up model size. We further elucidate the design space of language models for vision generation, including tokenizer choice, model choice, model scalability, vocabulary design, and sampling strategy through extensive comparative experiments. Our work is the first to analyze the optimization behavior of language models in vision generation, and we believe it can inspire more effective designs when applying LMs to other domains. Finally, our elucidated language model for image generation, termed as **ELM**, achieves state-of-the-art performance on the ImageNet 256×256 benchmark.

# 1 INTRODUCTION

In the domain of artificial intelligence generated content (AIGC), text and image generation (Brown, 2020; Ho et al., 2022) represent the principal focal points. Despite their shared goal of content generation, these two modalities predominantly employ distinct modeling methods. On the one hand, text generation is commonly facilitated by autoregressive (AR) language models, like LLaMA-3 (Touvron et al., 2023a) and GPT-4 (Achiam et al., 2023), which operate by predicting subsequent tokens based on preceding ones in a sequence. On the other hand, image generation predominantly utilizes diffusion models, such as Dall·E 3 (Betker et al., 2023) and Stable Diffusion v3 (Esser et al., 2024), which learn to gradually denoise images for all pixels simultaneously.

The recent success of large language models (LLMs) has bolstered the research community's confidence in their potential contribution towards achieving artificial general intelligence (AGI). This optimism has also inspired researchers within the computer vision domain to extend the AR paradigm to applications beyond text, such as image generation (Esser et al., 2021; Yu et al., 2021; Tian et al., 2024) and video generation (Kondratyuk et al., 2024). Such explorations open up novel avenues for leveraging AR in visual content creation. A significant advantage of integrating LLMs into image generation is the ability to transfer established techniques from text-based applications, such as generating content that exceeds the input length. In contrast, diffusion models generally exhibit less flexibility in adapting to such capabilities. Moreover, the scalability of LLMs makes them the preferred foundation for building unified models with multi-modal inference capabilities (Team et al., 2023; Kondratyuk et al., 2024). Gaining a deeper understanding of their potential across domains will aid the community in building more efficient and effective universal models.

Nevertheless, current research in visual generation remains focused on diffusion models (Karras et al., 2022; Ma et al., 2023; Kingma & Gao, 2024), and language models for image generation have yet to be thoroughly explored. Current efforts (Esser et al., 2021; Chang et al., 2022; Yu et al., 2022; Sun et al., 2024; Tian et al., 2024; Yu et al., 2024) are mostly preliminary, involving the discretization of images into sequences of tokens with vector-quantization autoencoders, which are then processed by language models trained with token prediction objectives. However, considering that text and images represent fundamentally different modalities, it is essential to thoroughly analyze the training dynamics and elucidate the design space when adapting LLM for image generation tasks.

In this study, we delve into the potential of language models for vision generation tasks. We quantitatively analyze the fundamental differences between images and text and conduct a comprehensive exploration of the design space for image generation using language models. Starting with image tokenization, we compare two approaches: VQGAN, which uses a vector quantizer (VQ) to discretize latent codes (Van Den Oord et al., 2017; Esser et al., 2021), and BAE, which employs binary autoencoders for "look-up free" quantization (LFQ) (Wang et al., 2023; Yu et al., 2023). Our comparison based on reconstruction ability, scalability, and generation performance shows that BAE consistently outperforms VQGAN across all dimensions. Despite this, current language model-based image generation methods largely rely on vector-quantization auto-encoders (Yu et al., 2022; Chang et al., 2022; Li et al., 2023; Sun et al., 2024). We believe that a more powerful quantizer for images can lead to significantly better generation performance. We then evaluate the performance of two primary language modeling approaches for image generation: autoregressive (AR) models and masked language models (MLMs). Consistent with findings in the language domain (Henighan et al., 2020; Liao et al., 2020; Zhang et al., 2024; Chang & Bergen, 2024), AR models demonstrate superior image generation ability and scalability compared to MLMs. We further leveraged the flexibility of the binary-valued bit codes produced by BAE. Through our exploration of code decomposition strategies, we found that splitting the original code into two subcodes significantly reduces learning complexity, improves performance, and reduces computational costs.

Additionally, we analyze how AR models learn to generate images by examining attention scores across different layers and model sizes. Our findings indicate that AR models effectively learn the importance of local information for image generation. However, larger models also capture global information, which is more difficult for smaller models to learn, helping to explain the performance improvements observed with increasing model size. Our research deepens the understanding of the LLM's capability and behavior in vision generation. The insights can contribute to the design of more efficient and unified large models handling multi-modalities inference tasks and the exploration of general artificial intelligence systems. In conclusion, our main contributions include:

- We identify the fundamental differences between the token distributions of discretized images and text, highlighting significant disparities in training dynamics and terminal phases between them.

- We thoroughly examine two prevalent language modeling methods, including AR models and MLMs, within the realm of image generation. Our findings suggest that AR mechanism holds greater potential in the visual domain.

- Leveraging an image discretization mechanism with BAE, our results reveal that a vocabulary decomposition helps improve performance and reduce computational cost.

- We show that AR models can learn effective image patterns without inductive bias, identify distinct patterns across model sizes, and offer a concise explanation of the scaling law.

- Combining all key ingredients of the design space explicitly explored, we reach a strong **E**lucidated **L**anguage model for i**M**age generation, termed as **ELM**, and achieve state-of-the-art performance on the ImageNet 256×256 benchmark.

## 2 PRELIMINARY

### 2.1 IMAGE TOKENIZATION

Image tokenization typically involves an encoder ENC, a quantizer QUANT, and a decoder DEC. Given an image $\boldsymbol{x} \in \mathbb{R}^{H \times W \times 3}$, ENC encodes it to latent variables $\boldsymbol{z} = \text{ENC}(\boldsymbol{x}) \in \mathbb{R}^{\frac{H}{f} \times \frac{W}{f} \times D}$, where $f$ is the down-sample factor and $D$ is the latent dimension. Each spatial vector $\boldsymbol{z}_{ij}$ in $\boldsymbol{z}$ is then quantized to discrete code $\boldsymbol{q}_k$. Let the quantized latent be denoted as $\boldsymbol{z}_q$, which is then decoded to reconstruct the original image as $\hat{\boldsymbol{x}} = \text{DEC}(\boldsymbol{z}_q)$ (Van Den Oord et al., 2017; Razavi et al., 2019; Esser et al., 2021; Yu et al., 2023). All the codes form a codebook $\mathcal{Q} = \{\boldsymbol{q}_k\}_{k=1}^K \subset \mathbb{R}^D$ that contains $K$ codes in total. The codebook can be viewed as the "vocabulary" if we regard the image as a special kind of language. A sequence of tokens $\boldsymbol{q} = (\boldsymbol{q}_1, \boldsymbol{q}_2, ..., \boldsymbol{q}_L)$, where $L = \frac{H}{f} \times \frac{W}{f}$, is obtained by reshaping $\boldsymbol{z}_q$ to a sequence of $L$ tokens.

**VQGAN (Esser et al., 2021)** For this method, the codebook $\mathcal{Q}$ is trained alongside the encoder and decoder, the most widely used one named VQGAN. In this method, each spatial latent vector $\boldsymbol{z}_{ij} \in \mathbb{R}^D$ "looks up" the nearest code $\boldsymbol{q}_k$ by minimizing the Euclidean distance:

$$\boldsymbol{z}_q = \text{QUANT}(\boldsymbol{z}) := \left( \arg\min_{\boldsymbol{q}_k \in \mathcal{Q}} \|\boldsymbol{z}_{ij} - \boldsymbol{q}_k\| \right) \in \mathbb{R}^{\frac{H}{f} \times \frac{W}{f} \times D}. \tag{1}$$

**BAE (Wang et al., 2023; Yu et al., 2023)** This method discretizes the scalar value at each position of the latent vector, converting it to a binary value (0/1 or -1/1) (Fajtl et al., 2020; Wang et al., 2023; Yu et al., 2023). Specifically, suppose the latent vector $\boldsymbol{z}_{ij} \in \mathbb{R}^D$ is normalized and the values lie within the range of (0,1). Each value $z^d, d \in \{1, ..., D\}$ at the $d$-th position of $\boldsymbol{z}_{ij}$ is further quantized into discrete values of 0 or 1:

$$z_q^d = \text{sign}(z^d) = \begin{cases} 0, & \text{if } z^d < 0.5, \\ 1, & \text{otherwise.} \end{cases} \tag{2}$$

In this way, the codebook is structured within a binary latent space, with $K = 2^D$. The code index is derived by treating the code as a binary number and converting it into its corresponding decimal value; this method is also referred to as "look-up free" quantization (LFQ) (Yu et al., 2023). The sign function can be replaced by Bernoulli sampling, then $\boldsymbol{z}_q = \text{Bernoulli}(\boldsymbol{z})$ (Wang et al., 2023).

### 2.2 MODELING METHODS

**Autoregressive (AR) Model** Consider a sequence of discrete tokens $\boldsymbol{q} = (\boldsymbol{q}_1, \boldsymbol{q}_2, ..., \boldsymbol{q}_L)$, where each token $\boldsymbol{q}_l$ is drawn from a vocabulary $\mathcal{Q}$ of size $K$. The AR model assumes that the probability of the current token $\boldsymbol{q}_l$ depends only on its preceding tokens $(\boldsymbol{q}_1, \boldsymbol{q}_2, ..., \boldsymbol{q}_{l-1})$, framing the generation task as a 'next-token' prediction, using unidirectional attention with the transformer architecture. Specifically, the network learns the probability $p(\boldsymbol{q}) = \Pi_{l=1}^L p(\boldsymbol{q}_l \mid \boldsymbol{q}_1, \cdots, \boldsymbol{q}_{l-1})$, with the loss function:

$$\mathcal{L}_{\text{ar}} = -\mathbb{E}_{\boldsymbol{x} \sim p(\boldsymbol{x})} [\log p(\boldsymbol{q})] \tag{3}$$

**Masked Language Model (MLM)**    Unlike AR models, MLMs leverage contexts from both directions to predict the tokens masked by a special [MASK] token, and their predictions are not bound by sequential order. They are trained by substituting a subset of tokens with [MASK] tokens and then predicting these tokens based on the unmasked ones. Specifically, There exists a binary mask $\boldsymbol{m} = [m_l]_l^L$ where the token $\boldsymbol{q}_l$ is replaced with [MASK] if $m_i = 1$, otherwise, when $m_i = 1$ will be left intact. Denote $\boldsymbol{q}_M$ the result after applying mask $\boldsymbol{m}$ to $\boldsymbol{q}$. Hence, these models optimize the following loss function:

$$\mathcal{L}_{\mathrm{mlm}} = -\mathbb{E}_{\boldsymbol{x} \sim p(\boldsymbol{x})} \left[ \sum_{\forall l \in [0,L], \ m_l = 1} \log p(\boldsymbol{q}_l \mid \boldsymbol{q}_M) \right] \tag{4}$$

As a dominant modeling approach in the language domain, there have been several attempts to adapt AR transformer models for image synthesis (Esser et al., 2021; Yu et al., 2022; Team, 2024; Sun et al., 2024). At the same time, MLMs also gain popularity in the vision domain due to their sampling efficiency (Chang et al., 2022; Li et al., 2023; Chang et al., 2023).

## 3    ELUCIDATING THE DESIGN SPACE OF LANGUAGE MODELS FOR IMAGE GENERATION

In this section, we first analyze the intrinsic difference between vision and language domain based on the token distribution, which helps us to understand the learning behavior of language models on the image generation task. Then we comprehensively explore the design space of adopting language models for vision generation, including the tokenizer choice, modeling choice, model scalability analysis, vocabulary decomposition strategy with BAE tokenizer, and sampling strategy.

### 3.1    IMAGE GENERATION VERSUS TEXT GENERATION

While images can be discretized and treated as token sequences, the inherent differences between vision and text still exist. These disparities result in varying performance while both are trained using the same model architectures and objectives. In our experiments, we observe that the training loss did not converge well using either AR or MLM on image tokens, a similar result is also presented in Henighan et al. (2020). However, the models can still generate high-quality images with a low Fréchet Inception Distance (FID) (Heusel et al., 2017), indicating that they have learned sufficient patterns for image generation, although the training loss remains high.

Table 1: KL-divergence between token distribution and Uniform Distribution, along with the perplexity of n-gram models.

| | **ImageNet** | | | | **OpenWebText** | | **WallStreetJournal** | |
|---|---|---|---|---|---|---|---|---|
| Tokenizer | VQGAN-f16 (V=16384) | | BAE-f16 (V=65536) | | BPE (V=47589) | | BPE (V=19979) | |
| | unigram | bigram | unigram | bigram | unigram | bigram | unigram | bigram |
| Train | 1.00 | 2.16 | 0.24 | 0.17 | 3.25 | 3.35 | - | - |
| Val | 0.90 | 2.12 | 0.22 | 0.03 | 3.27 | 1.94 | - | - |
| Perplexity [1] | 368 | 210 [2] | 52,538 | 596,855 | 2087 | 395 | 962 | 170 |

**Token Distribution and Randomness in Image Data**    Our analysis shows that image tokens exhibit a distribution much closer to a random, uniform distribution when compared to language tokens, and they exhibit a lack of orderliness based on bigram distribution and n-gram models' perplexity (see the result in **Table** 1). These observations lead to several key implications. First, it

---

[1] We calculate the perplexity with Laplace smoothing(Gale & Church, 1994). The first 10 percent of the training data is select the efficiently calculate the perplexity of OpenWebText.

[2] Although the VQGAN tokenizer exhibits lower perplexity compared to BAE, its extremely low code utilization significantly impacting the tokenizer's effectiveness.

suggests that *image data lacks the inherent structure and sequential order* typically present in language data, implying that image generation is less dependent on strict sequential patterns and more on *local patterns* relevant to visual reconstruction (Ulyanov et al., 2018). Second, a token distribution close to uniform highlights that the generation task has a *higher tolerance for errors*. Since all tokens are nearly equally probable, the model can afford to make less precise token predictions without significantly impacting the quality of the generated output. This characteristic explains why our model, despite its high training loss, can still generate high-quality images—a behavior consistent with prior findings on deep learning's robustness to unstructured data (Zhang et al., 2016; Arpit et al., 2017). The principles of information theory (Shannon, 1948) also point out that models dealing with more random data require less precision in capturing global relationships.

## 3.2 TOKENIZER CHOICE: VQGAN VERSUS BAE

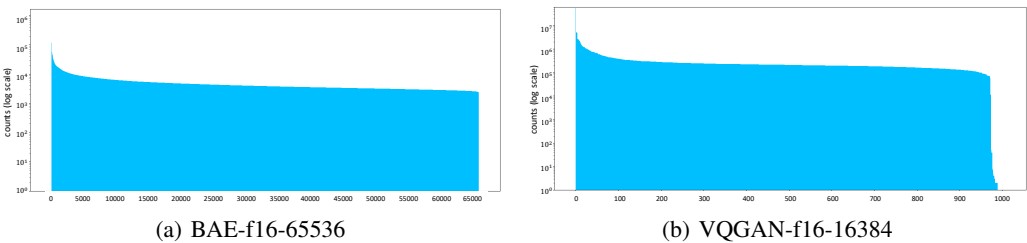

(a) BAE-f16-65536        (b) VQGAN-f16-16384

Figure 2: **BAE-16 exhibits a higher code utilization than VQGAN-f16.** This figure shows a log count number of the appearance of codes on the ImageNet training dataset in sorted order. (a) BAE-16, with a code dimension of 16, has 65,536 unique codes and achieves 100% code utilization, with no code showing extremely low usage. In contrast, (b) VQGAN-f16, with a codebook size of 16,384, only utilizes around 1,000 codes, and many of these codes have extremely low utilization.

In VQGAN, "code collapse" is a critical issue where a large portion of the codebook remains unused as the codebook size increases, severely limiting the model's efficiency and scalability Zhu et al. (2024); Baykal et al. (2024). This problem does not occur in BAEs, where discrete codes are generated using scalar quantization (Mentzer et al., 2023). This approach guarantees 100% code utilization (see **Figure** 2) and achieves better reconstruction capabilities (**Appendix** A.3). *Based on the above reasons, we build our generation model on BAE tokenizer instead of VQGAN.*

For BAE, we observe that the introduction of *Bernoulli Sampling* during quantization improves image generation performance (**Table** 8). Incorporating this probabilistic element reduces the model's sensitivity to prediction errors (Englesson & Azizpour, 2021), leading to a more robust generation.

## 3.3 MODELING METHOD CHOICE: AR VERSUS MLM

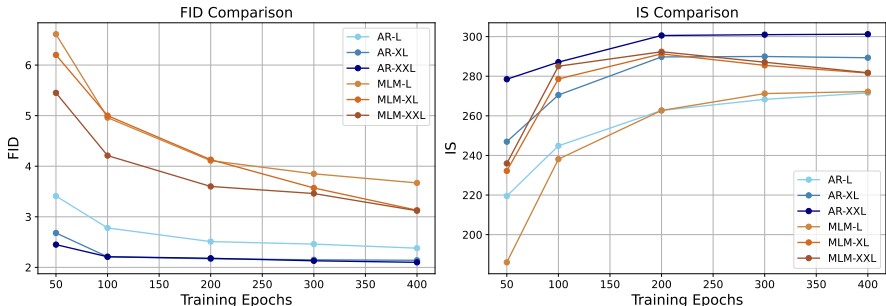

Figure 3: **Comparison of AR and MLM on image generation with 50,000 generated samples.** AR consistently outperforms MLM across various model sizes.

In this subsection, we evaluate the performance of both vanilla AR and MLM in image generation with the same BAE-f16 tokenizer with a vocabulary size of $2^{16}$ and training strategy (see implemen-

tation details in **Appendix** A.4). **Figure** 3 presents the FID score and Inception Score (IS) (Salimans et al., 2016) on the 256×256 ImageNet benchmark over the training epochs for both AR and MLM. The results show that AR consistently outperforms MLM across various model sizes. Additionally, AR exhibits higher training efficiency compared to MLM, particularly as the model size increases. Research in the language domain has widely recognized that AR models possess greater generative capabilities than MLMs, particularly as model scales increase (Radford et al., 2019; Raffel et al., 2020; Henighan et al., 2020). Our findings align with these research works. Besides, for MLM-XL and MLM-XXL, a clear divergence between FID and IS is observed in the later stages of training, where FID continues to improve, while IS declines. Studies point out that when models overfit to generate highly realistic samples (low FID), they may sacrifice diversity, which negatively impacts IS (Chong & Forsyth, 2020; Benny et al., 2021). This issue does not occur with AR models, further highlighting the superiority of AR models over MLMs in maintaining both quality and diversity.

### 3.4 LEARNING AND SCALING BEHAVIOR

To further understand the model's learned patterns, we visualize the attention maps of different AR models. These visualizations revealed that the attention mechanism were primarily focused on *local regions* of the image, indicating that the AR transformer models effectively learn the importance of local patterns for image generation (Vaswani et al., 2017). This finding is notable because the model was trained without any inductive biases tailored to image data, highlighting the strong capability of AR transformer models across different domains.

Additionally, the *scaling law* (Henighan et al., 2020; Kaplan et al., 2020) holds for AR models in image generation tasks, as reflected in *lower training loss* (**Figure** 14), *improved generation performance*, and an enhanced ability to capture *global information* as the model size increases. As for the attention pattern, models of varying sizes showed subtle differences: the L-sized model mainly focused on local information, struggling to capture long-term information. In contrast, larger models (XL and XXL) exhibited longer-range attention in certain layers, suggesting they had also learned global features (**Figure** 4). Specifically, in the first layer (layer 0), attention generally captures global information, while deeper layers show a more *localized focus*, with recent tokens receiving greater attention. In XL and XXL models, which have more layers, some deeper layers still capture global information. However, in L-sized models, the deeper layers also focus on local tokens, with little attention to long-term dependencies. The incorporation of global information positively impacted overall generation performance, as evidenced by the lower FID score and better visual image quality as shown in **Figure** 8.

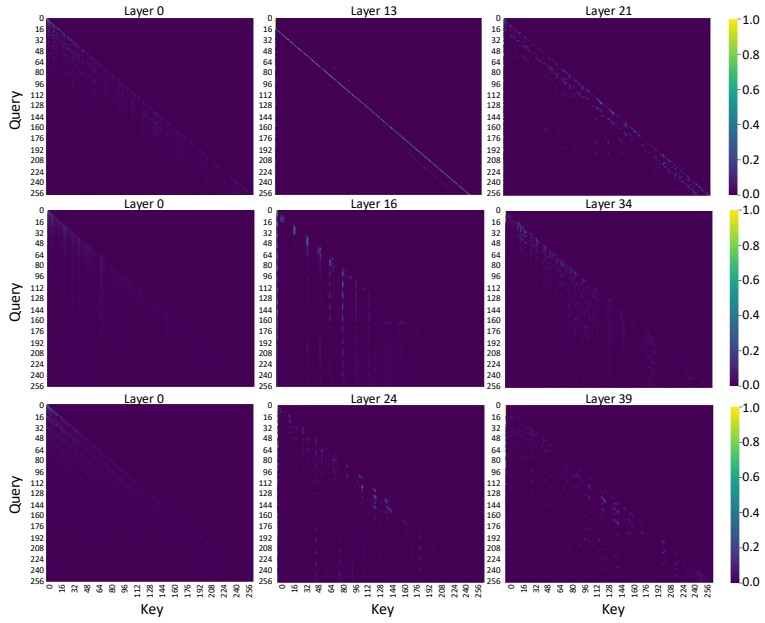

Figure 4: Visualization of average attention score of head 0 in AR models over 100 images. From the top row to the bottom row, we show the results of L, XL, and XXL models with the BAE 2-10 tokenizer, respectively. All models effectively learned to focus on *localized information* across different layers. However, larger model learns to capture richer *global information*, a behavior rarely observed in the L-sized models.

### 3.5 VOCABULARY DESIGN

The vocabulary size $K$ in BAE tokenizer is determined by the code dimension $D$, *i.e.*, $K = 2^D$. However, when the vocabulary size exceeds a certain threshold, such as $2^{16}$ (*i.e.*, 65,536), next-token prediction becomes significantly more challenging (Ali et al., 2023). For even larger vocabulary sizes, such as those exceeding $2^{20}$, it becomes infeasible due to memory constraints. Despite these limitations, the tokenizer's effectiveness largely depends on the code dimension, as demonstrated by the results in **Figure** 5, which shows that increasing the

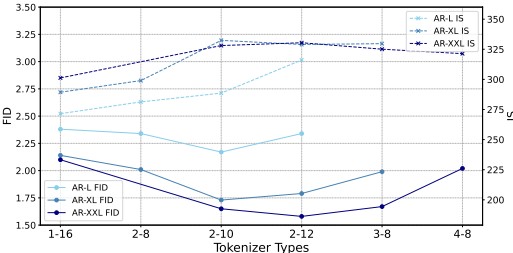

Figure 5: AR model performance with different BAE tokenizers.

code dimension improves reconstruction ability. Recent research also indicates that a stronger tokenizer leads to a better generation performance in AR models (Tao et al., 2024).

To address the challenge of large vocabulary sizes, we leverage the flexibility of binary-quantized codes that allows us to decompose each code into multiple subcodes (Yu et al., 2023). For instance, an 8-bit code like $[1, 0, 1, 0, 0, 0, 1, 1]$ can be split into two 4-bit codes: $[1, 0, 1, 0]$ and $[0, 0, 1, 1]$. These two subcodes can then be converted into decimal values to generate corresponding indices. As a result, we convert the embedding matrix from a size of $2^8 \times D_{feature}$ into two matrices of size $2^4 \times D_{feature}$, where $D_{feature}$ is the feature dimension within the AR model. The final embedding is achieved by concatenating the two indexed embeddings and applying a projection to restore the dimension to $D_{feature}$. Separate prediction heads are applied to generate the logits.

We conduct experiments using AR models with BAE that have varying code dimensions ($D = 16$, 20, 24, and 32). We treat quantizers with and without code decomposition as distinct tokenizers; for example, for $D = 16$, "1-16" means the original tokenizer and "2-8" denotes the code is split into two 8-bit subcodes. The results in **Figure** 5 reveal several key insights:

● **Optimal decomposition.** A decomposition into *two subcodes* is generally optimal, which also *reduces computational costs*, leading to more *efficient and effective generation* (see the detailed result in **Table** 10). When dealing with two sub-vocabularies of smaller size, the prediction at each position is split into two independent classification tasks, each with a more manageable set of possible outcomes, largely reducing the cognitive load on the model (Ali et al., 2023; Yang, 2024). Further increasing the number of subcodes significantly raises the prediction complexity, and the model struggles to optimize across three or more dimensions (Limisiewicz et al., 2023). This is evidenced by the increasing training loss observed when moving from tokenizers 2-8 to 3-8 and 4-8 in XL and XXL models (see **Figure** A.4). The added complexity in managing multiple classification heads impairs the model's generalization, leading to suboptimal outcomes in image synthesis.

● **Vocabulary complexity and model capacity.** Larger code dimensions generally lead to improved generation performance but introduce more complex vocabularies, making it harder for the model to predict the next token. As a result, *more complex tokenizers require more powerful models* for effective learning (Tao et al., 2024). For example, the 2-10 tokenizer is optimal for L and XL models, while the 2-12 tokenizer performs best with the XXL model.

These findings demonstrate the trade-offs between model scale, vocabulary complexity, and decomposition strategies, highlighting the potential of the AR model's ability to effectively handle complex tokenization while maintaining high performance across model scales.

### 3.6 SAMPLING STRATEGY

Sampling strategy plays a crucial role in vision generation, applicable to both diffusion models (Karras et al., 2022; Ma et al., 2023) and language models (Chang et al., 2022; Sun et al., 2024). In this study, we thoroughly explore the sampling strategies for both AR and MLM, including classifier-free guidance (Ho & Salimans, 2022) (CFG) scale, the introduction of randomness, and the number of generation iterations for the MLMs.

Firstly, regarding the CFG scale, we discover that a gradually increasing CFG scale performs better than a constant one. We test various CFG scale scheduling methods (as illustrated in **Figure** 16) and find that linear scheduling yields the best results (see the result in **Table** 11).

Secondly, regarding the introduction of randomness, for the AR model, randomness primarily derives from the $k$ value in the top-$k$ filter used when selecting next-token indices based on their confidence scores; a larger $k$ introduces more randomness. For the MLM, randomness mainly stems from the coefficient $\tau$ of Gumbel noise added to the confidence of the [MASK] token predictions; a larger $\tau$ results in greater randomness. We observed that for both methods, a high degree of randomness is crucial during the sampling process (see **Figure** 6, 7 and **Table** 12). This finding is consistent with the natural randomness of image token distribution discussed in **Section** 3.1. Moreover, as model size and vocabulary increase, the need for randomness diminishes, indicating that larger models are capable of capturing a *broader range of patterns* and making *more accurate predictions*. This observation aligns with the attention and scalability analysis discussed earlier, where larger models demonstrated enhanced capacity to manage both local and global information, reducing the need for stochasticity to generate realistic samples.

For MLMs, the range of sampling iterations during generation varies from 1 to the total sequence length (i.e., $16 \times 16 = 256$). We conclude that the optimal number of iterations is around 10 (**Figure** 17), which reflects the sampling efficiency of MLMs compared to AR models.

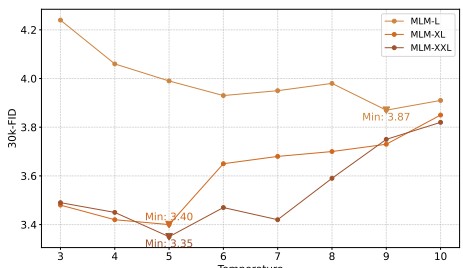
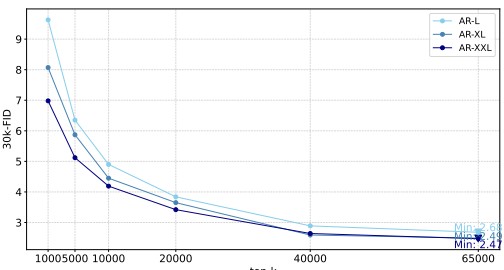

Figure 6: The results of different $\tau$ with MLM.    Figure 7: The results of different top-$k$ with AR.

### 3.7 ELM MODEL

After thoroughly exploring the design space of language models for image generation, via combining the better design trick, we reach our final **E**lucidated **L**anguage model for i**M**age generation (**ELM**). ELM adopts BAE as the image tokenizer and AR as the modeling method. According to our previous results, ELM splits the quantized image code into two subcodes. When choosing the vocabulary, the capacity of the model should be considered. Larger vocabularies require more powerful models to handle next-token prediction (2-12 tokenizer works best for ELM-XXL), while smaller models perform better with simpler vocabularies (ELM-L and XL perform better with 2-10 tokenizer). For sampling strategy, we choose a *high randomness* because it will bring in large diversity, and we use linear CFG. Finally, we construct four ELM versions: ELM-L (2-10), ELM-XL (2-10), ELM-XXL (2-12), and ELM-2B (2-12), with parameters ranging from 315M to 1.9B.

## 4 EXPERIMENTS

### 4.1 CONDITIONAL IMAGE GENERATION

In this section, we compare our ELM models with other AR models for image generation. Our experiments are conducted on the $256 \times 256$ ImageNet (Deng et al., 2009) dataset. We generate 50,000 samples to evaluate performance using FID, IS, Precision, and Recall following Dhariwal & Nichol (2021). Implementation details can be found in the **Appendix** A.4. The comparison result is presented in **Table** 2. Our method (ELM) exhibits scaling law behavior, with performance improving as model size increases, also achieves state-of-the-art (SOTA) results. Given that our tokenizer (BAE) is only trained on ImageNet (Deng et al., 2009), we believe further training on larger datasets, like OpenImages (Kuznetsova et al., 2020), would enhance the tokenizer and further boost the generation capability of our ELMs.

Table 2: Comparison of AR models on class-conditional image generation on 256×256 ImageNet. * indicates that the model generates samples at a resolution of 384×384, which are then resized to 256×256. -re denotes rejection sampling is used.

| Type | Model | Params. | FID↓ | IS↑ | Precision↑ | Recall↑ |
|------|-------|---------|------|-----|-----------|---------|
| **Diff.** | DiT-L/2 (Peebles & Xie, 2023) | 458M | 5.02 | 167.2 | - | - |
| | DiT-XL/2 | 675M | 2.27 | 278.2 | 0.83 | 0.57 |
| | SiT-XL/2 (ODE) (Ma et al., 2024) | 675M | 2.15 | 258.1 | 0.81 | 0.60 |
| | SiT-XL/2 (SDE) | 675M | 2.06 | 277.5 | 0.83 | 0.59 |
| **MLM** | MaskGIT | 227M | 6.18 | 182.1 | 0.8 | 0.51 |
| | MaskGIT-re | 227M | 4.02 | 355.6 | - | - |
| **AR** | VQGAN (Esser et al., 2021) | 227M | 18.65 | 80.4 | 0.78 | 0.26 |
| | VQGAN-re | 1.4B | 5.20 | 280.3 | - | - |
| | LlamaGen-L (Sun et al., 2024) | 343M | 3.81 | 248.3 | 0.83 | 0.52 |
| | LlamaGen-XL | 775M | 3.39 | 227.08 | 0.81 | 0.54 |
| | LlamaGen-XXL | 1.4B | 3.09 | 253.61 | 0.83 | 0.53 |
| | LlamaGen-3B | 3.1B | 3.05 | 222.33 | 0.80 | 0.58 |
| | LlamaGen-3B* | 3.1B | 2.18 | 263.33 | 0.81 | 0.58 |
| **VAR** | VAR-d16 (Tian et al., 2024) | 310M | 3.30 | 274.4 | 0.84 | 0.51 |
| | VAR-d20 | 600M | 2.57 | 302.6 | 0.83 | 0.56 |
| | VAR-d24 | 1.0B | 2.09 | 312.9 | 0.82 | 0.59 |
| | VAR-d30 | 2.0B | 1.97 | 334.7 | 0.81 | 0.61 |
| | VAR-d30-re | 2.0B | 1.80 | **356.40** | 0.83 | 0.57 |
| **MAR** | MAR-B (Li et al., 2024) | 208M | 2.31 | 281.7 | 0.82 | 0.57 |
| | MAR-L | 479M | 1.78 | 296.0 | 0.81 | 0.60 |
| | MAR-H | 943M | 1.55 | 303.7 | 0.81 | 0.62 |
| **AR** | **ELM**-L (2-10) | 315M | 2.17 | 288.59 | 0.82 | 0.55 |
| | **ELM**-XL (2-10) | 757M | 1.79 | 332.99 | 0.80 | 0.59 |
| | **ELM**-XXL (2-12) | 1.4B | 1.58 | 330.43 | 0.80 | 0.60 |
| | **ELM**-2B (2-12) | 1.9B | **1.54** | 332.69 | 0.81 | 0.60 |

## 4.2 VISUALIZATION OF THE SCALING LAW

According to the scaling law of ELM transformers, both loss and performance improve as training data and model parameter size increase. In our experiments, although the original image data (ImageNet) remains unchanged, the token set effectively scales up through the token decomposition strategy. We present generated samples using different sizes of ELM models (L, XL, XXL) and tokenizers (1-16, 2-10, 2-12) to illustrate the scaling behavior of ELM models in image generation. Following Tian et al. (2024), we maintain the same seed and teacher-forced initial tokens across models. The results in **Figure** 8 clearly demonstrate performance improvements as both the token set and model size scale up.

## 5 RELATED WORK

**Large Language Models.** Language models are foundational tools in natural language processing, designed to predict the likelihood of sequences of words or tokens, using Transformer architectures with self-attention mechanism (Vaswani et al., 2017). There are two primary types: autoregressive (AR) models, like GPT (Radford et al., 2019; Brown, 2020; Achiam et al., 2023), LLaMA (Touvron et al., 2023a;b; Dubey et al., 2024), etc., which generate text one token at a time in a left-to-right fashion, and masked language models (MLM), such as BERT (Devlin, 2018), T5 (Raffel et al., 2020), etc., which predict masked tokens within a sequence using bidirectional context. AR models are particularly effective for text generation due to their sequential nature, while MLMs are better suited for representation learning by leveraging global context (Chang & Bergen, 2024). The scaling law (Henighan et al., 2020; Kaplan et al., 2020), which describes the relationship between the growth of model parameters, dataset sizes, computational resources, and performance improvements, highlights the immense potential of AR models.

**Vision Generation.** Vision generation is a key focus in the current AIGC field, primarily relying on diffusion probabilistic models, which generate images by progressively denoising a random Gaus-

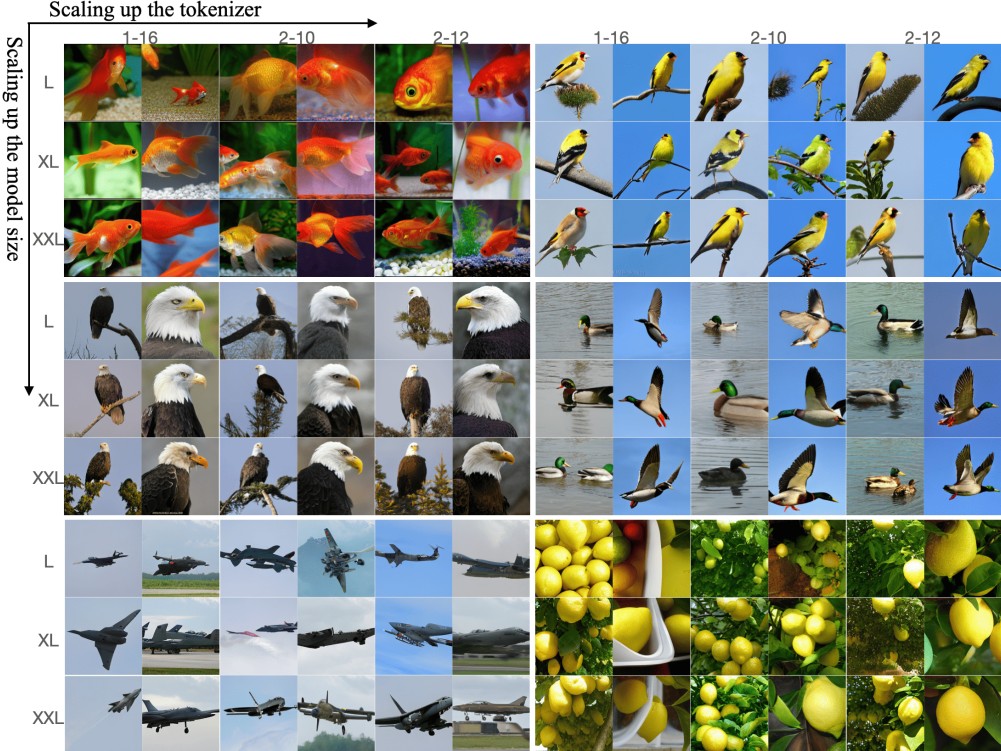

Figure 8: Scaling up behavior of tokenizer and model size. From left to right and top to bottom, there is a trend of improved image detail and structure. It reflects the enhanced generation ability that comes with more refined tokenizer and larger model.

sian noise (Song et al., 2020; Peebles & Xie, 2023; Chen et al., 2023). Transformer architectures are also the dominant backbone in these tasks. Language models have also been applied to vision tasks. Researches like Chang et al. (2022), Li et al. (2023), and Chang et al. (2023) use bidirectional MLMs for image generation, meanwhile Esser et al. (2021), Yu et al. (2022), Sun et al. (2024), and Tian et al. (2024) employ AR models. Moreover, AR models offer a path toward developing unified models for general artificial intelligence across different modalities, as seen in systems like Gemini (Team et al., 2023) and Chameleon (Team, 2024). While previous research has explored the use of language models in vision generation, our work is the first to analyze fundamental differences between text and image and the optimization behavior of language models in vision domain.

## 6 CONCLUSION

In this work, we investigate the use of language models for image generation. We analyze the differences between image and text token distribution, demonstrating how these distinctions affect training behavior, and offering insights that extend beyond current research on language models for image generation. We further elucidate the design space of language models for vision generation, including tokenizer choice, model choice, model scalability, vocabulary design, and sampling strategy through extensive comparative experiments. Through our analysis, we have the following findings: (1) binary autoencoder (BAE) demonstrates superior performance as an image tokenizer compared to traditional VQGAN approaches; (2) AR models consistently outperform MLMs and show a strong scaling law, (3) larger vocabulary size and a decomposition design benefit the image generation, (4) sampling strategies should also allow for *greater randomness*; gradually increased CFG scale, larger top-$k$ are important for a better FID score. By combining these designs, we reach our final ELM model, and it achieves state-of-the-art performance on ImageNet. We hope this work will motivate further usage of the AR model across other domains.

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

## A  APPENDIX

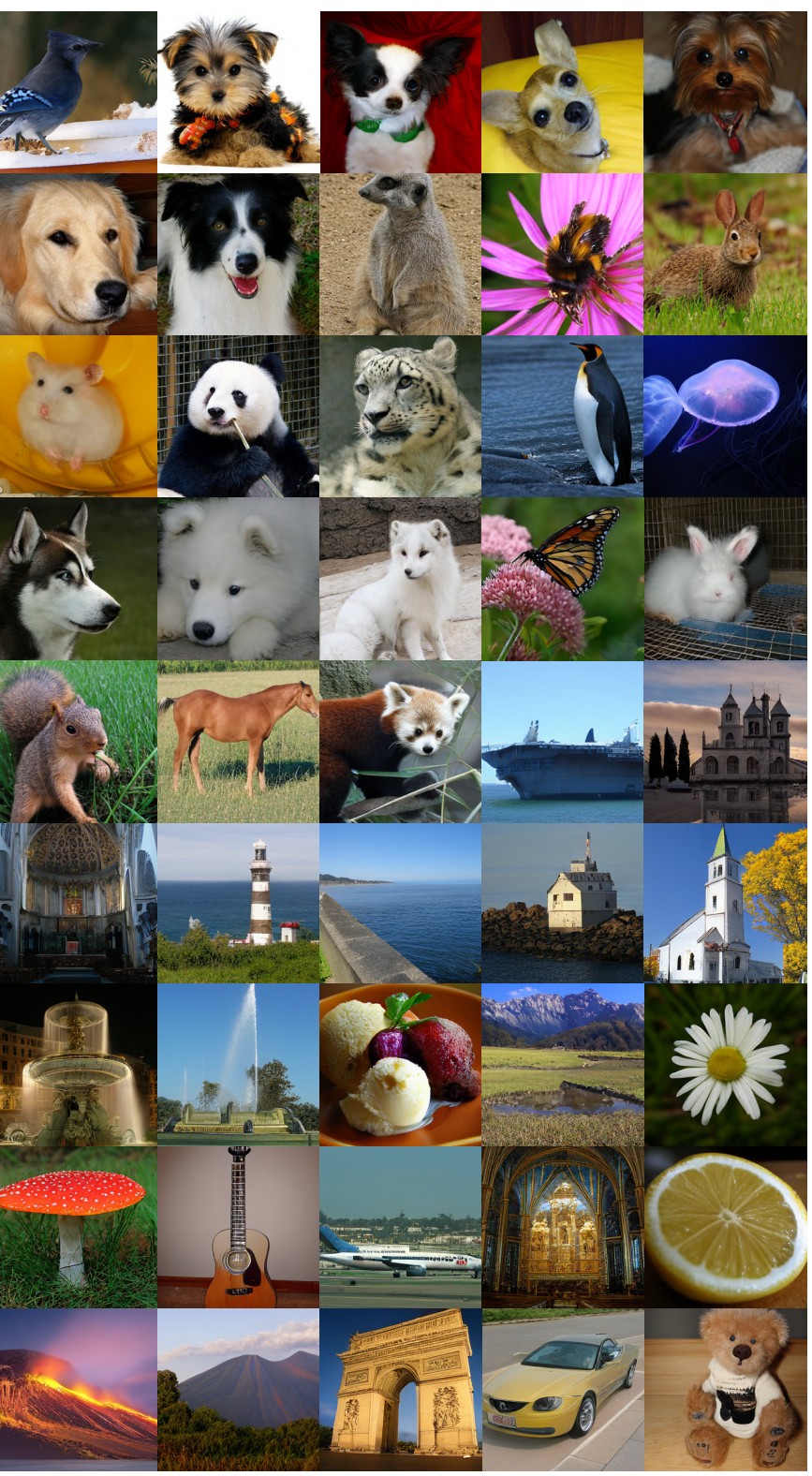

Figure 9: Selected samples in different classes with ELM-2B (2-12).

## A.1 ADDITIONAL ROBUSTNESS ANALYSIS OF ELM MODELS

We conduct additional experiments to demonstrate the robustness of the elucidated language models in the vision domain, including zero-shot generalization, higher-resolution generation, and evaluations on different datasets.

### A.1.1 PERFORMANCE ON ZERO-SHOT GENERALIZATION

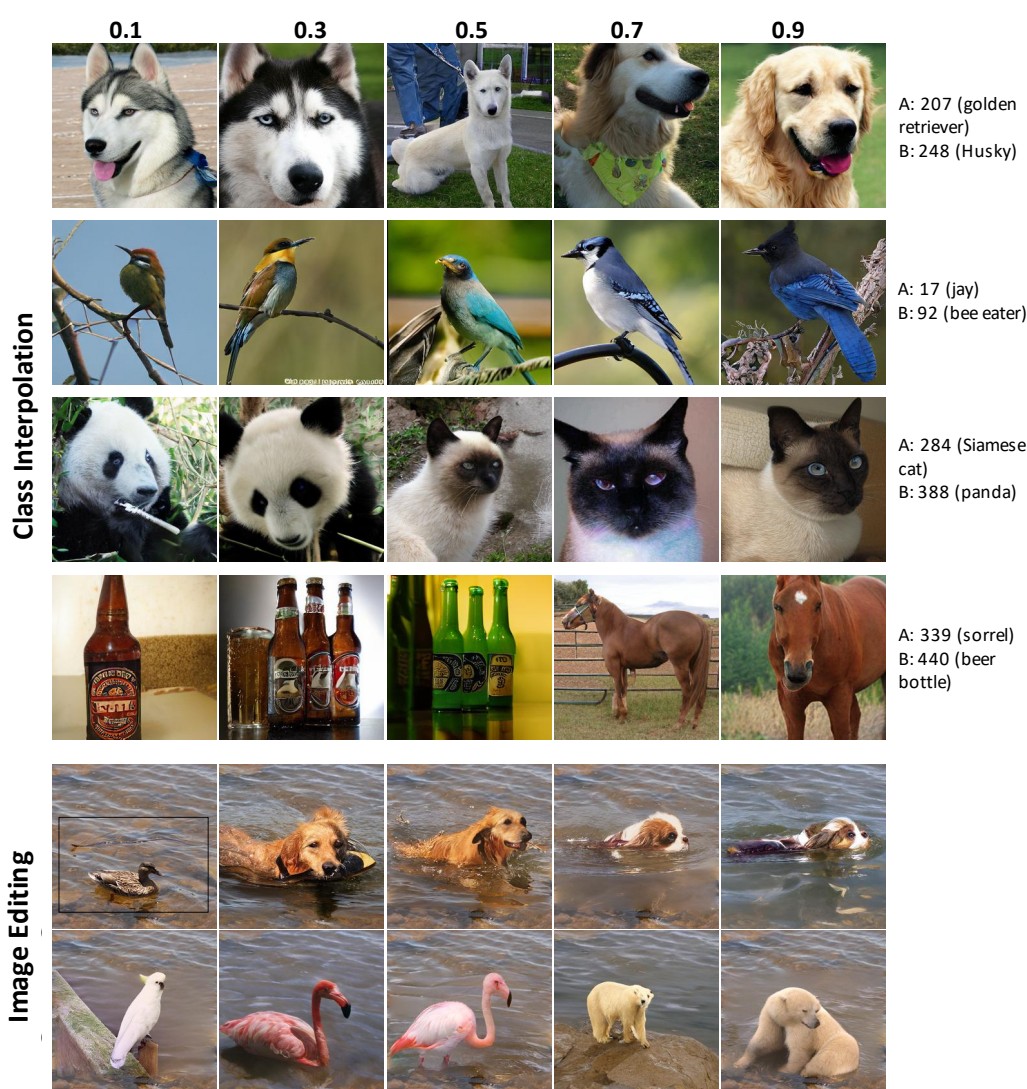

Figure 10: Zero-shot generalization performance of ELM. Class interpolation generate images with interpolated class condition, i.e., $\alpha A + (1-\alpha)B, \alpha \in \{0.1, 0.3, 0.5, 0.7, 0.9\}$. Image editing allows the model to edit the masked region based on specific class condition.

We evaluated the model's performance on generating images with interpolated class conditions, specifically, $\alpha A + (1-\alpha)B$, where $A$ and $B$ are two distinct class labels, $\alpha \in [0, 1]$. This approach effectively tests how the model learns and adapts to conditions, especially under complex scenarios. The results show that the model effectively learns the conditional information, rather than simply memorizing it. Interestingly, when the interpolated classes share similarities, such as a golden retriever and a husky, the model generates images that blend features of both classes when $\alpha$ is around

0.5. In contrast, for unrelated classes like a sorrel and a beer bottle, the generated images only reflect the features of the class with the higher weight. The image editing results further highlight the flexibility of ELM across various application tasks.

### A.1.2 PERFORMANCE ON HIGHER RESOLUTION

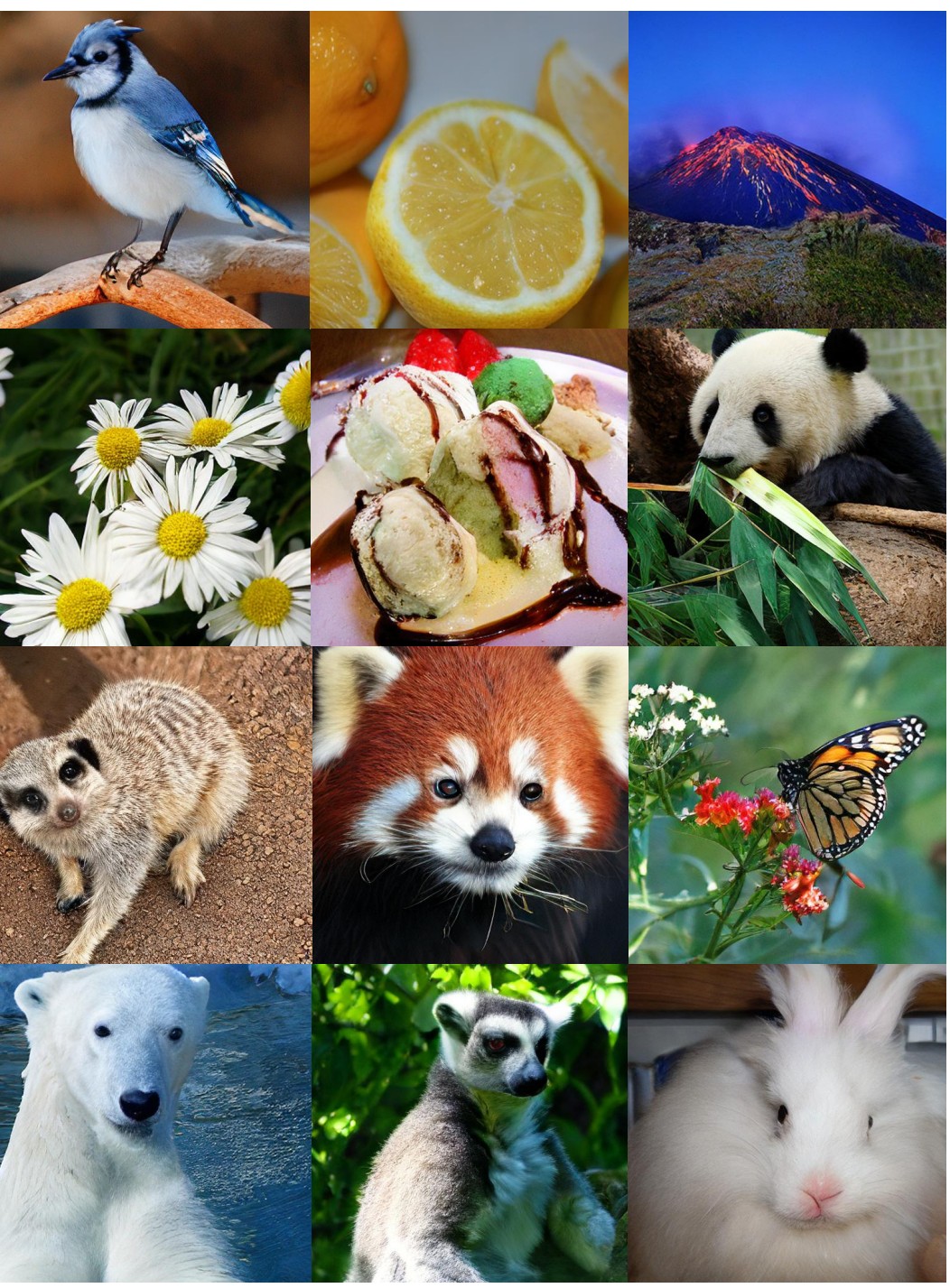

Figure 11: Generated 512×512 images.

Table 3: **Comparisons on class-conditional ImageNet 512×512 benchmark.**

| Model | Tokenizer | Params. | train. steps | FID↓ | IS↑ | Precision↑ | Recall↑ |
|-------|-----------|---------|--------------|------|-----|-----------|---------|
| DiT-XL/2 | VAE | 675M | 3000k | 3.04 | 240.82 | 0.84 | 0.54 |
| MaskGIT | VQGAN | 227M | 1500k | 7.32 | 156.0 | 0.78 | 0.50 |
| ELM-L | BAE 2-8 | 312M | 250k | 4.82 | 246.87 | 0.81 | 0.59 |

To showcase the versatility of the elucidated model, we conducted experiments on higher-resolution datasets. Specifically, we trained an ELM-L with a 2-8 tokenizer on 512x512 ImageNet. The model was initialized with parameters from a version pretrained on 256x256 datasets and further trained for only 50 epochs (250,000 training iterations with 256 batch size). The selected images and the quantitative results presented in **Table** 3 demonstrate ELM's potential on higher-resolution datasets. This approach also offers a more applicable way to train image generation models for high resolutions, as training directly on higher resolutions can be challenging and the data is often scarce. However, with ELM models, we can start training on abundant data at lower resolutions, and the model can be easily adapted to higher resolutions without any modifications.

### A.1.3 PERFORMANCE ON DIFFERENT DATASETS

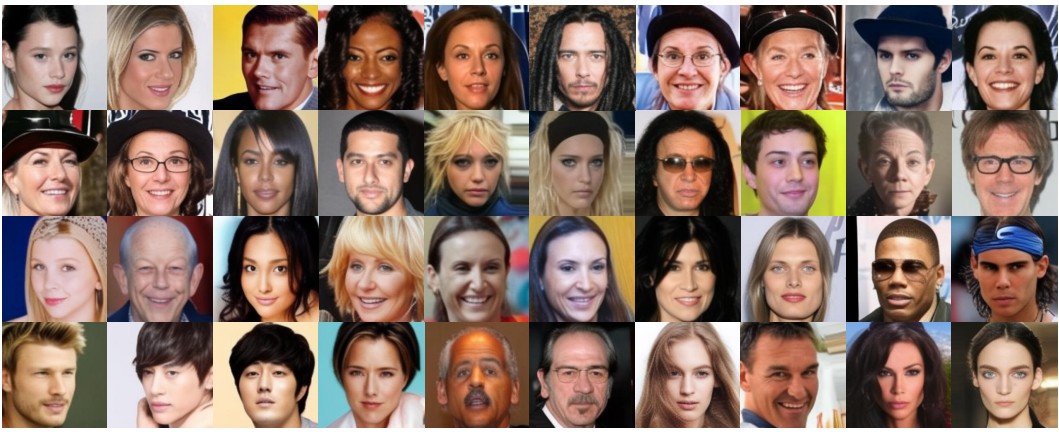

Figure 12: Generated human face images with 256×256.

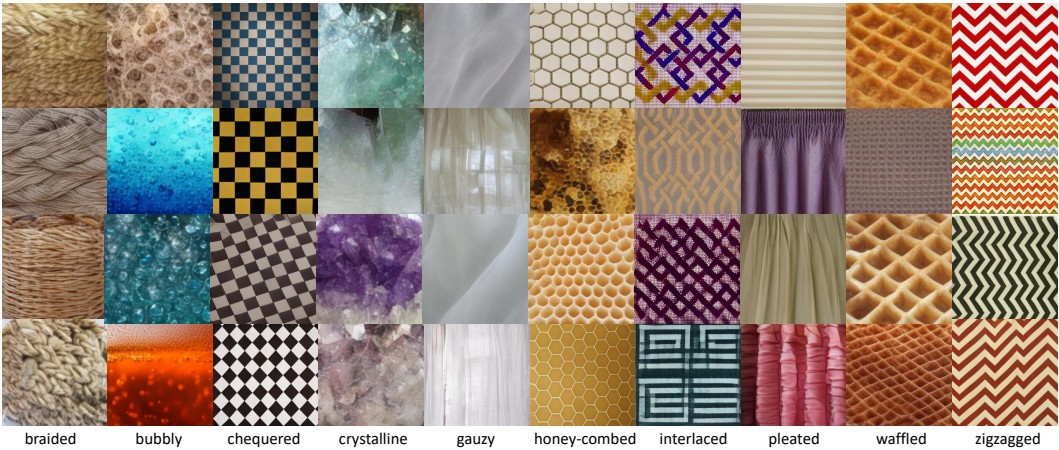

braided    bubbly    chequered    crystalline    gauzy    honey-combed    interlaced    pleated    waffled    zigzagged

Figure 13: Generated special texture images with 256×256.

We conduct experiments on specialized datasets distinct from ImageNet to assess the robustness and versatility of ELM models. Specifically, we select CelebA (Liu et al., 2015), which includes 202,599 human face images across 10,177 identities, and the Describable Texture Dataset (DTD) (Cimpoi et al., 2014) that comprises 5,640 images across 47 different categories. We train an ELM-L model with a 2-8 tokenizer on each dataset for 400 epochs using a batch size of 256. The qualitative results (**Figure** 12 and 13) from these experiments demonstrate the high performance of our model across diverse types of tasks.

## A.2 INTRINSIC DIFFERENCE BETWEEN LANGUAGE AND IMAGES

We choose ImageNet from the image domain; OpenWebText and Shakespeare[3] from the language. The information of the tokenized dataset is shown in **Table** 4 and the KL-divergence between uniform distribution is shown in **Table** 1.

We can see that from **Table** 1, compared to text generation, image generation exhibits a higher randomness. Note that although VQGAN-f16 generates tokens with a lower level of randomness, the major reason is the low code utilization-only less than 10% code from the vocabulary is used, and the generated image quality is not satisfying due to the extremely low token utilization.

Table 4: Vocabulary (Codebook) information of image and text.

|  | **ImageNet** |  | **OpenWebText** | **WallStreetJournal**[4] |
|---|---|---|---|---|
| Tokenizer | VQGAN-f16 | BAE-f16 | BPE | BPE |
| Vocab size | 16384 | 65536 | 47589 | 19979 |
| Token num of train set | 327M | 327M | 9B | 38M |
| Token num of val set | 12M | 12M | 4M | 1.5M |

## A.3 COMPARISON BETWEEN BAE AND VQVAE

We trained BAE on the ImageNet dataset using the same model architecture and loss functions as VQGAN from the taming transformers framework (Esser et al., 2021). For a fair comparison, we evaluated the VQGAN-f16-16384 model[5] that also trained on the ImageNet dataset, and assessed its code utilization (see **Figure** 2. The results clearly demonstrate that BAE outperforms VQGAN, achieving lower reconstruction FID (rFID) (**Table** 5) and generation FID (gFID) (**Table** 6) and significantly higher code utilization (100% v.s. 8%).

Table 5: **Reconstruction FID of the image tokenizers.** All tokenizer are trained on the ImageNet. * indicates the value is directly copy form https://github.com/CompVis/taming-transformers.

|  | **VQGAN**-f16 |  | **BAE**-f16 |  |  |  |
|---|---|---|---|---|---|---|
| codebook size | 1024 | 16384 | $2^{16}$ | $2^{20}$ | $2^{24}$ | $2^{32}$ |
| rFID | 10.54* | 7.41* | 3.32 | 2.24 | 1.77 | 1.68 |

## A.4 ADDITIONAL EXPERIMENT RESULTS

**Implementation Details**  For the BAE tokenizer, we followed the configuration in Wang et al. (2023), utilizing Bernoulli sampling during quantization, and trained it for 400 epochs on the ImageNet dataset. For the transformer model, we adopted the LLaMA-2 (Touvron et al., 2023b) architecture, as referenced in Sun et al. (2024). The depth and feature dimensions of each model size are detailed in **Table** 7. All language models were trained on 80GB A800 GPUs with a batch size of 256, for 400 epochs, using a constant learning rate of 1e-4, weight decay of 0.05, and the

---

[3]Obtainted from https://github.com/karpathy/nanoGPT

[4]The information is obtained from Standford lecture note: https://web.stanford.edu/ jurafsky/slp3/3.pdf

[5]Downloaded from https://github.com/CompVis/taming-transformers

Table 6: **Generation FID of AR-L with different image tokenizers.** AR model is trained on the ImageNet for 1,000,000 iterations, 200 epochs. We generate 30,000 samples for each model. 'cfg1-3' denotes classifier-free guidance (cfg) scale gradually increased to 3.0 following a linear schedule across inference iteration. 'cfg1.5' denotes the cfg remains fixed at 1.5 during inference.

| tokenizer | code dim | vocab. size & top-$k$ | cfg1-3 gFID | cfg1.5 gFID | rFID |
|---|---|---|---|---|---|
| **VQGAN**-f16 | 256 | 16,384 | 6.71 | 8.12 | 7.41 |
| **BAE**-f16 | 16 | 65,536 | 2.78 | 3.87 | 3.32 |

AdamW optimizer with $\beta_1$ 0.9 and $\beta_2$ 0.95. The L and XL-sized models were trained on 8 A800 GPUs, requiring approximately 6.4 and 10 days, respectively, to complete 400 epochs. The XXL-sized model, trained on 16 A800 GPUs (2 nodes with 8 GPUs each), took around 12 days to finish training.

For the AR model, we implement mainly follow Sun et al. (2024), except for the 2B-sized model. The MLM and AR models use the same model architecture. For the MLMs training strategy, we mainly follow Chang et al. (2022). Specifically, at each training step, we sample a mask ratio for each sample, mask tokens based on this ratio, and train the model to predict the masked tokens. The mask ratio follows a cosine schedule across the generation iterations, meaning the process transitions from less to more information. Early in training, most tokens are masked; as training progresses, the mask ratio sharply decreases, revealing more tokens for the model to handle in later stages.

Table 7: Transformer model architecture information with different sizes.

| Size | depth | dimension | num of head |
|---|---|---|---|
| ELM-L | 24 | 1024 | 16 |
| ELM-XL | 36 | 1280 | 20 |
| ELM-XXL | 48 | 1536 | 24 |
| ELM-2B | 48 | 1792 | 28 |

Table 8: **The influence of Bernoulli sampling with BAE on FID (30k) of generation.** We test on AR-L model with BAE-f16 with $D = 16$, and the model is trained for 150 epochs.

| cfg | constant 2 | linear1-3 |
|---|---|---|
| w. Bernoulli | 4.72 | 2.88 |
| w.o. Bernoulli | 5.05 | 3.13 |

**Comparison of Tokenization w. and w.o Bernoulli Sampling**  When using BAE to tokenize image feature codes into discrete tokens, the process can either be deterministic, by directly converting values to 0 or 1 based on a threshold, or nondeterministic by incorporating Bernoulli sampling during quantization. We compared both methods to assess their impact on the generation task. As shown in **Table** 8, the nondeterministic approach clearly performs better. This result aligns with the inherent randomness of image token distribution, as discussed in **Section** 3.1, and offers greater tolerance for classification errors during next-token prediction.

**Comparison Between AR Model and MLM**  Table 9 shows the detailed final result of the different-sized AR models and MLMs using the basic BAE-f16 on the ImageNet 256×256 dataset. Clearly, AR models always show better performance than MLMs.

**Scaling Behavior of AR Models**  Figure 14 show the loss trends of all sized AR models (L, XL, XXL and 2B) with 2-12 tokenizer. All models successfully converged, and the final loss consistently decreased as model size increased.

Table 9: **Comparison of AR and MLM on ImageNet 256×256**. The auto-encoder is BAE-f16 with code dimension 16. The FID results are obtained on 30K generation images.

| Size | Method | FID↓ | sFID↓ | IS↑ | Precision↑ | Recall↑ |
|------|--------|------|-------|------|-----------|---------|
| L    | MLM    | 3.67 | 5.34  | 272.23 | 0.8561 | 0.4597 |
|      | AR     | 2.38 | 4.78  | 271.54 | 0.8201 | 0.5650 |
| XL   | MLM    | 3.13 | 4.95  | 261.59 | 0.8159 | 0.5355 |
|      | AR     | 2.14 | 4.92  | 289.33 | 0.8162 | 0.5834 |
| XXL  | MLM    | 3.12 | 4.86  | 281.75 | 0.8393 | 0.4947 |
|      | AR     | 2.10 | 4.89  | 301.22 | 0.8284 | 0.5839 |

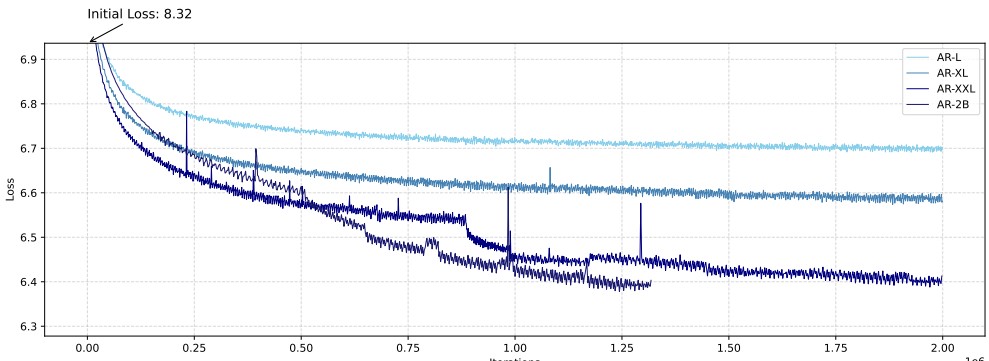

Figure 14: **AR exhibits a good scaling law.** Training losses of all AR models are with the BAE 2-12 tokenizer. All models were trained for 2,000,000 iterations, equivalent to 400 epochs, except the 2B model, which had to be stopped earlier due to time constraints.

**Comparison Between Code-decomposition Strategies** **Table** 10 shows the detailed results of AR models with different BAE tokenizers. The code decomposition strategy significantly influences the model parameter size and the generation performance. For the code decomposition strategy, splitting a large vocabulary into **two** smaller sub-vocabularies yields optimal performance by *balancing vocabulary size with the number of classification heads*. In general, larger code dimensions improve generation performance by offering finer granularity, but they also introduce more complex vocabularies, making it increasingly challenging for the model to predict the next token accurately.

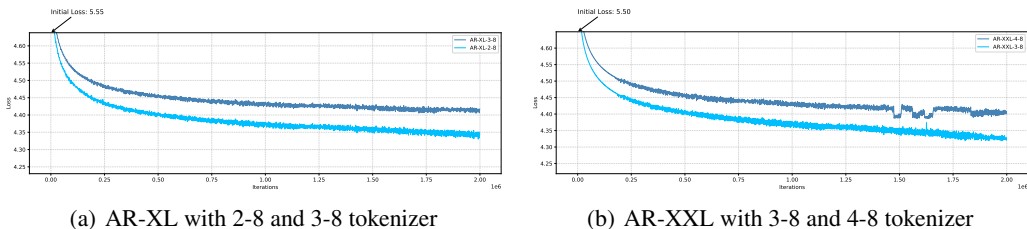

(a) AR-XL with 2-8 and 3-8 tokenizer          (b) AR-XXL with 3-8 and 4-8 tokenizer

Figure 15: **Different vocabulary decomposition strategies vary a lot on the training losses.** Clearly, introducing more than two classification heads will increase the model's training complexity and learning effectiveness.

**Comparison Between Sampling Strategies** For MLMs, we conduct a search to find the optimal CFG scale, iteration number, and temperature $\tau$ for the Gumbel noise (see **Figure** 6). For AR models, we search for the best CFG scale and top-$k$ threshold. During the search process, we

Table 10: **Comparisons of AR models on class-conditional ImageNet 256×256 benchmark.**

| Model | Tokenizer | Params. | FID↓ | sFID↓ | IS↑ | Precision↑ | Recall↑ |
|-------|-----------|---------|------|-------|-----|------------|---------|
| L | 1-16 | 443M | 2.38 | **4.78** | 271.54 | 0.8201 | 0.565 |
| | 2-8 | 312M | 2.34 | 4.86 | 281.29 | 0.8190 | 0.5573 |
| | 2-10 | 316M | **2.17** | 4.83 | 288.59 | 0.8168 | 0.5536 |
| | 2-12 | 328M | 2.34 | 5.12 | **316.08** | 0.8197 | 0.5487 |
| XL | 1-16 | 900M | 2.14 | 4.92 | 289.33 | 0.8162 | 0.5834 |
| | 2-8 | 737M | 2.01 | 4.50 | 298.99 | 0.8069 | 0.5979 |
| | 2-10 | 741M | **1.73** | **4.50** | **332.38** | 0.8183 | 0.5823 |
| | 2-12 | 757M | 1.79 | 4.82 | 328.99 | 0.8027 | 0.5903 |
| | 3-8 | 740M | 1.99 | 5.29 | 329.66 | 0.8070 | 0.5906 |
| XXL | 1-16 | 1.56B | 2.10 | 4.89 | 301.22 | 0.8284 | 0.5839 |
| | 2-10 | 1.37B | 1.65 | **4.33** | 328.08 | 0.8144 | 0.5933 |
| | 2-12 | 1.39B | **1.58** | 4.78 | **330.43** | 0.8034 | 0.6091 |
| | 3-8 | 1.37B | 1.67 | 4.99 | 325.06 | 0.8020 | 0.6054 |
| | 4-8 | 1.37B | 2.02 | 5.66 | 321.37 | 0.7913 | 0.602 |
| 2B | 2-12 | 1.90B | **1.54** | 4.81 | **332.69** | 0.8093 | 0.5968 |

calculate the FID score using only 30k samples for efficiency, noting that the FID values obtained in this way are consistently higher than those calculated with 50k samples.

Classifier-free guidance (CFG) plays a crucial role in conditional image generation, but it involves balancing the trade-off between image diversity and individual image quality. We searched for the optimal CFG scale for all models. Additionally, we found that using a dynamic CFG schedule significantly improves performance. We tested several CFG scheduling methods (see **Figure** 16), with the results summarized in **Table** 11.

Table 11: **Different CFG strategies varies a lot on FID.** All results are based on AR-L with tokenizer 2-10.

| CFG-scale | 1.5 | 2 | 2.5 | cos1-4 | log1-4 | linear1-4 | square1-4 | r-square1-4 |
|-----------|-----|---|-----|--------|--------|-----------|-----------|-------------|
| FID (30k) | 2.98 | 3.35 | 3.58 | 2.86 | 2.70 | **2.48** | 4.94 | 3.57 |

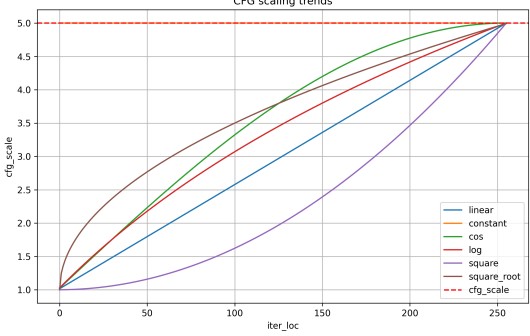

Figure 16: **Curves of CGF scale with respect to iteration times under different CFG schedules.**

### A.5   ELM IS FLEXIBLE TO GENERATE ANY-SIZE IMAGE

To further explore the capability of AR models in image generation, we generate images with more than 16×16 tokens without modifying the model (**Figure** 18). Although the model's receptive field

Table 12: **The influence of top-$k$ in sampling process** on 30k-FID scores for AR models with decomposed vocabulary.

|     | 2-8 | | | 2-10 | | | 2-12 | | |
|-----|------|------|------|------|------|------|------|------|------|
| $k$ | 180 | 210 | 256 | 800 | 900 | 1024 | 2600 | 2800 | 3000 |
| L   | 2.97 | 2.84 | **2.74** | 2.55 | 2.50 | **2.48** | 2.68 | **2.56** | 2.67 |
| XL  | 2.46 | **2.36** | 2.40 | 2.13 | 2.11 | **2.03** | 2.11 | **2.10** | 2.11 |
| XXL | -    | -    | -    | 2.08 | 2.04 | **1.95** | **1.90** | **1.90** | 1.95 |

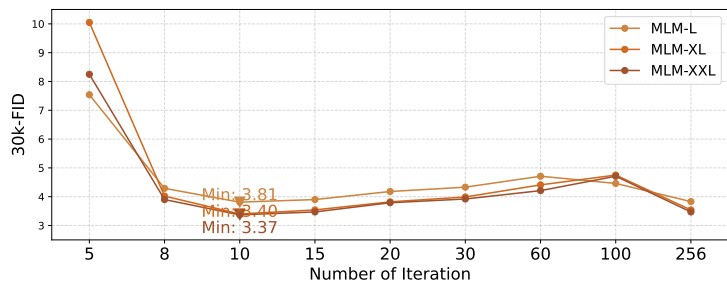

Figure 17: **The influence of iteration time (differnt mask ratio) in sampling process** on FID scores for MLMs.

Table 13: The best sampling strategy regards to FID score for all models.

| Method | Tokenizer | Model | Best Strategy |
|--------|-----------|-------|---------------|
| MLM | 1-16 | L | linear CFG 1-3; $\tau$=9.0, iteration number=10 |
|     | 1-16 | XL & XXL | linear CFG 1-3; $\tau$=5.0, iteration number=10 |
| AR | 1-16 | L& XL & XXL | linear CFG 1-3; top-$k$=65536 (all) |
|    | 2-8  | L | linear CFG 1-4; top-$k$=256 (all) |
|    | 2-8  | XL & XXL | linear CFG 1-4; top-$k$=210 |
|    | 2-10 | L | linear CFG 1-4; top-$k$=1024 (all) |
|    | 2-10 | XL & XXL | linear CFG 1-5; top-$k$=1024 (all) |
|    | 2-12 | L & XL & XXL | linear CFG 1-5; top-$k$=2800 |
|    | 3-8  | XL & XXL | linear CFG 1-5; top-$k$=180 |
|    | 4-8  | XXL | linear CFG 1-5; top-$k$=180 |

is limited to 256 tokens, we can easily generate 'streaming' images by looking back at a few tokens. This demonstrates the greater flexibility of AR models compared to diffusion models, highlighting the potential of AR models for applications in other domains.

## A.6 LIMITATION

Our work has limitations. While we propose several improvements for AR models, the fundamental issue of optimizing highly random token distributions remains. Traditional next-token prediction using classification loss may not be the most optimal training objective for such tasks, suggesting that more suitable objectives should be explored in future research. For instance, MAR (Li et al., 2024) has made promising progress by introducing diffusion loss into AR models, while VAR (Tian et al., 2024) presents a valuable perspective by altering the image tokenization approach. We hope our analysis will inspire further exploration and innovation in utilizing language models for vision generation, as well as other modalities.

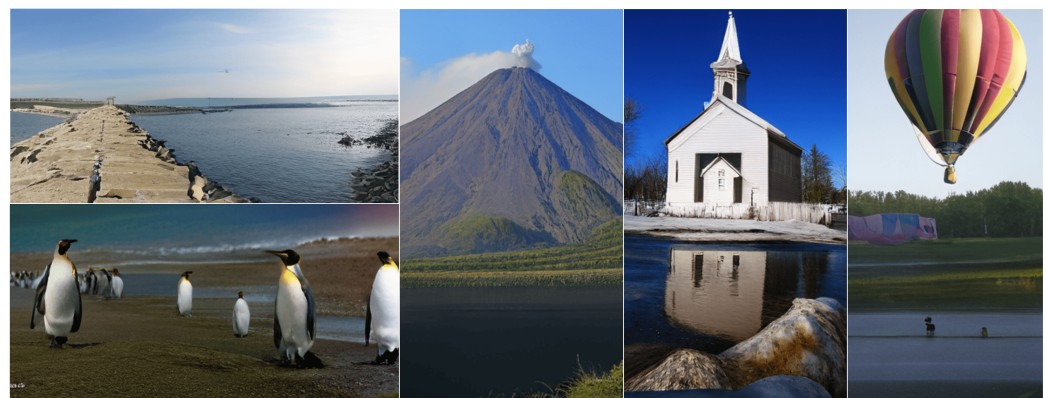

Figure 18: AR models are flexible to generate images with any size based on previous context.

## A.7 MORE GENERATED SAMPLES

We present more generated samples here to straightforwardly show the performance of our model.

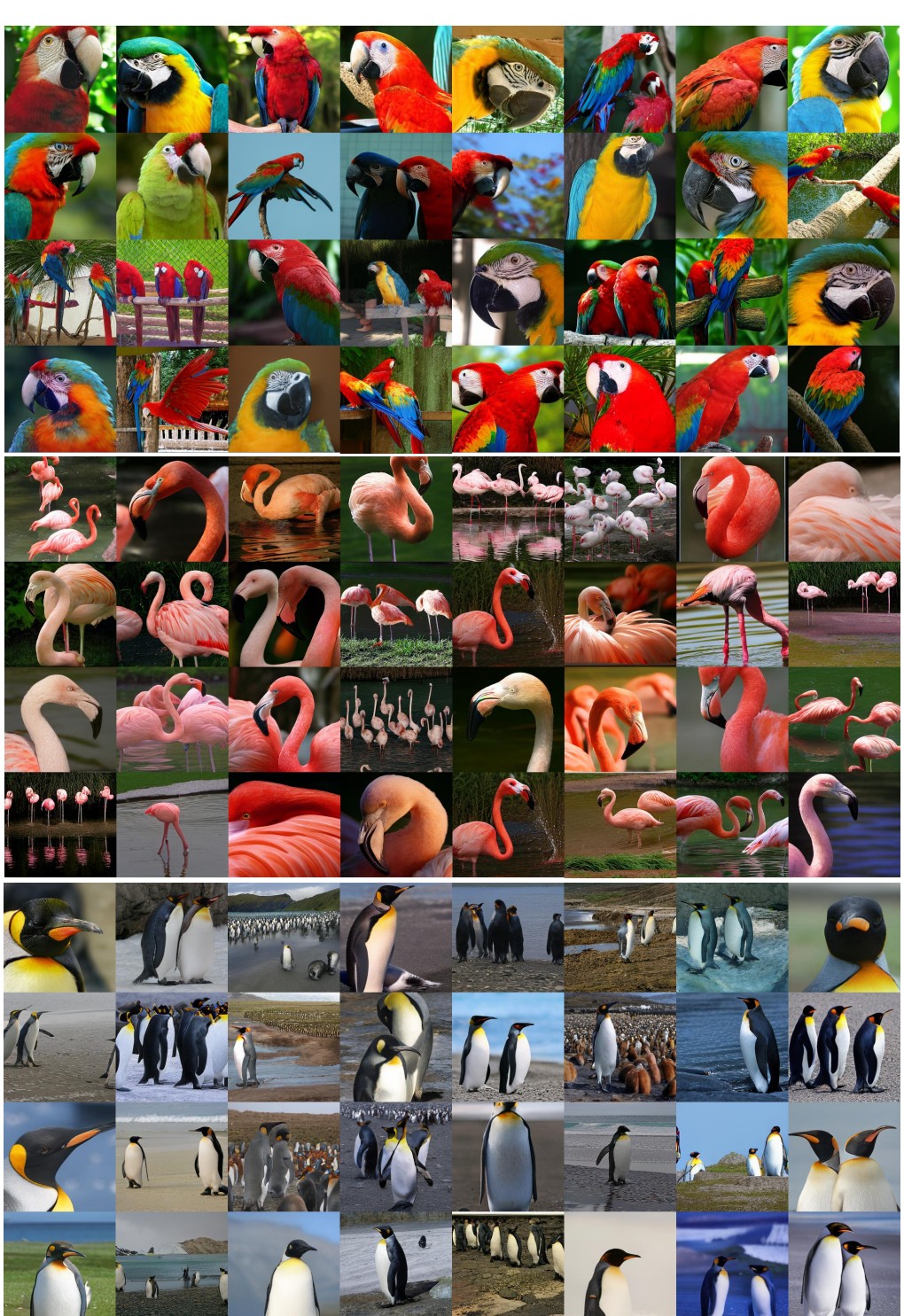

Figure 19: Randomly sampled images from classes 88 (macaw), 130 (flamingo), and 145 (king penguin).

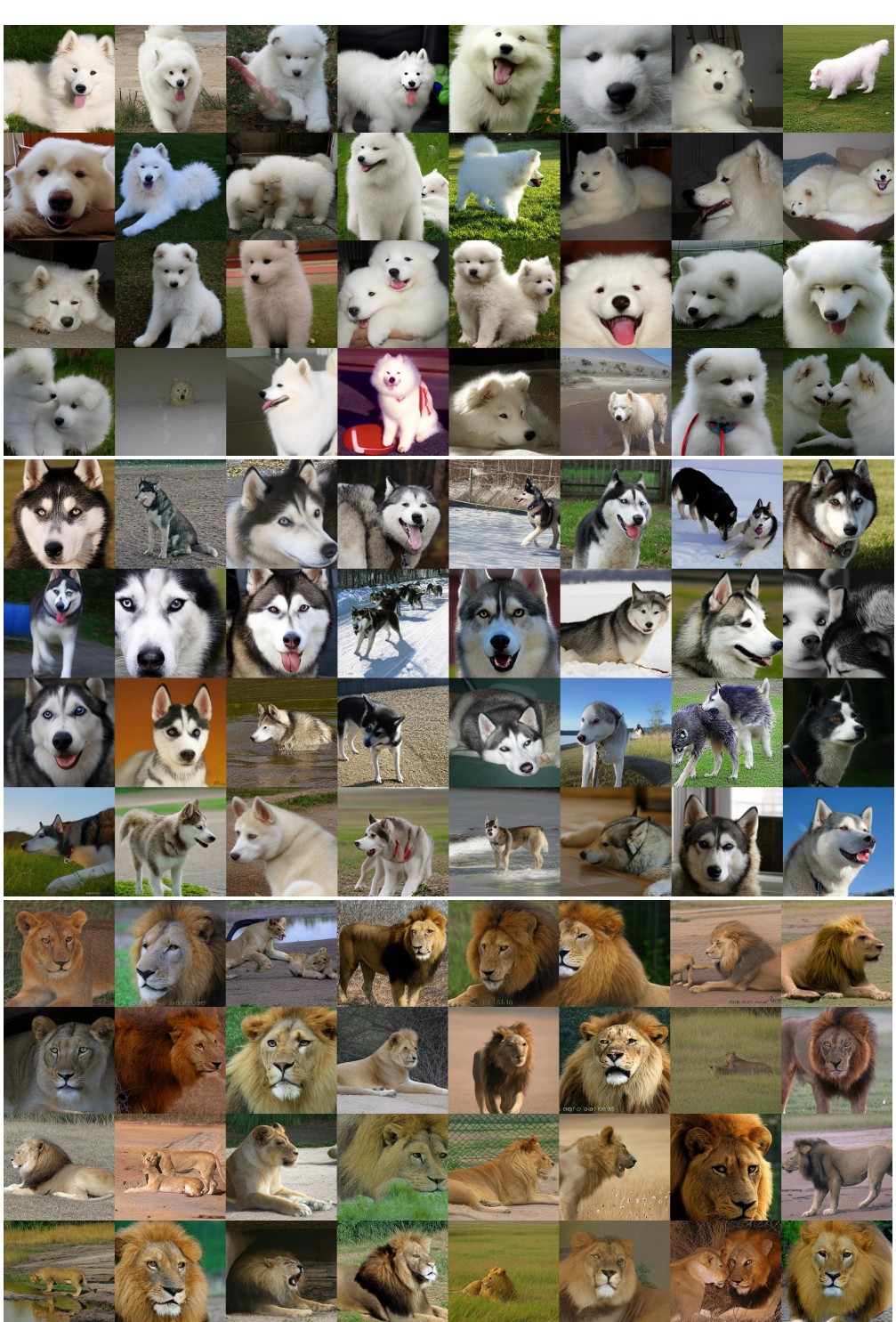

Figure 20: Randomly sampled images from classes 258 (Samoyed), 248 (Husky), and 291 (lion).

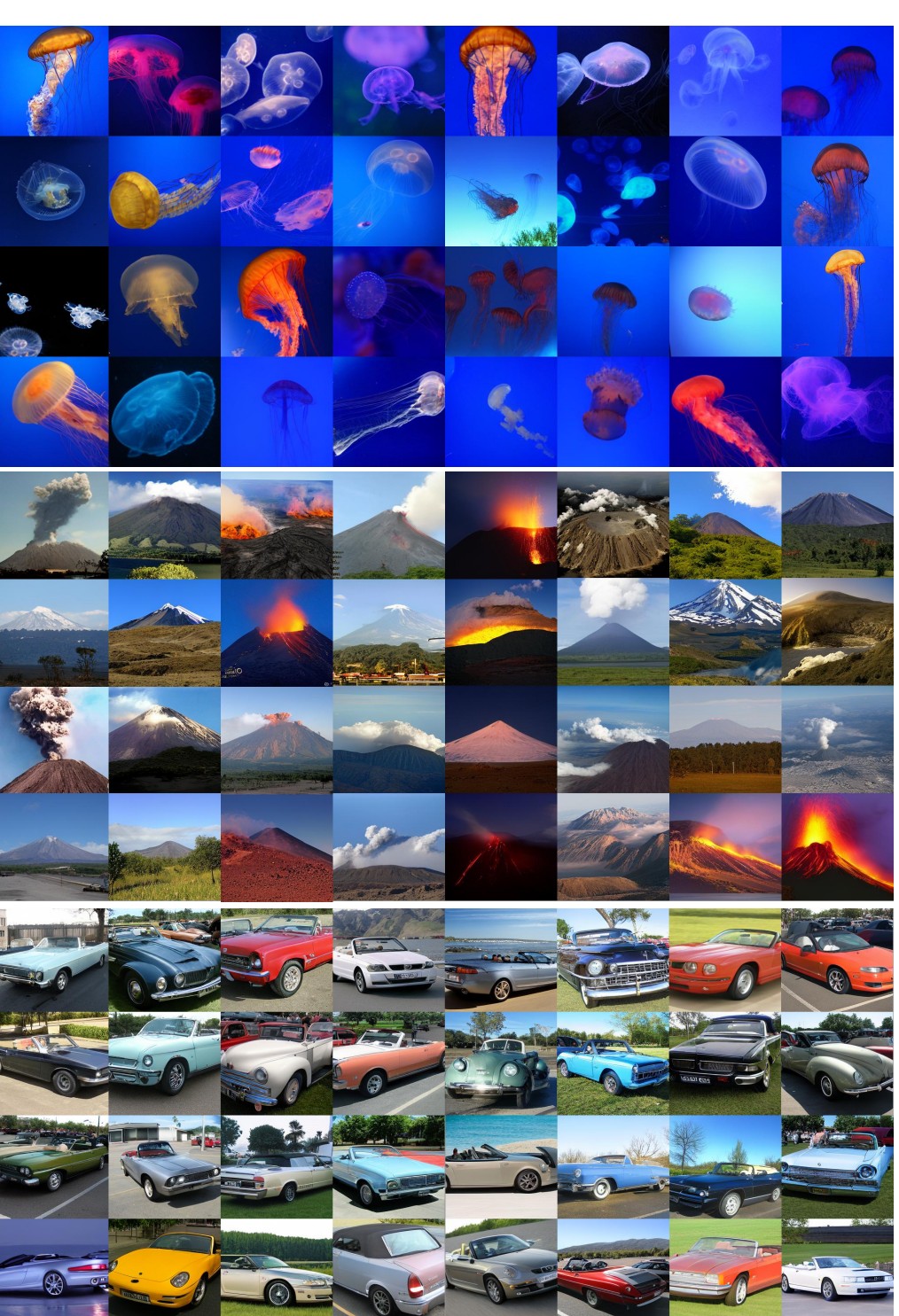

Figure 21: Randomly sampled images from classes 107 (jelly fish), 980 (volcano), and 511 (convertible).

