# OpenReview forum: "Elucidating the Design Space of Language Models for Image Generation"
_ICLR.cc/2025/Conference — Submitted to ICLR 2025_

### Official Review · Reviewer_kMTD · 2024-10-22

**Soundness:** 3
**Presentation:** 2
**Contribution:** 3
**Rating:** 5
**Confidence:** 3

**Summary:**

This explores the potential of adapting autoregressive (AR) language models for image generation tasks and study its design spaces. The study investigates how these models can be optimized for image generation, considering the differences between text and image data. The authors identify key challenges, such as the greater randomness in image tokens compared to text tokens, which complicates the training process.

A key focus of the paper is on analyzing the design space of language models for vision generation, including choices in tokenization (e.g., VQGAN vs. BAE), model scalability, and vocabulary design. Through extensive experiments, the authors find that BAE-based tokenization outperforms traditional vector-quantized approaches, and that AR models exhibit better scalability and image generation capabilities than masked language models (MLMs). The study also highlights the importance of model size, showing that larger AR models are better at capturing global context, while smaller models struggle with this aspect.

The main contribution is a comprehensive analysis of the design space for applying language models to image generation. The authors propose the Elucidated Language model for iMage generation (ELM), which achieves state-of-the-art performance on the ImageNet 256×256 benchmark. This work aims to inform future designs of language models for visual content creation and multi-modal inference.

**Strengths:**

The paper studies the fundamental differences between the token distributions of discretized images and text. Which is an interesting direction.

The paper demonstrate the effectiveness of AR models and its potential on image generation on  the ImageNet 256×256 benchmark.

**Weaknesses:**

The paper is not presented well and very empirical. I would suggest skip introducing too much details about preliminary works such as VQGAN, BAE without sharing insights designing them for image generation.

The main part of the paper is mainly about experimental comparisons but there are not comprehensive experiments made to support the claims.

Only image quality is reflected in figures such as Figure 8, I would suggest also adding text prompts and also possibly give more complex cases to compare the language understanding ability of autoregressively based LM for visual generation.

The fonts are too small in figure 4,5,6,7.

Only AR models are used as baselines in this paper.

**Questions:**

Overall, what do you think is the best strength of using LLM for visual generation than diffusion models?

For the tokenizer selection, have you considered using the VAE?

---

> ### Author Response · Authors · 2024-11-19
> **Official Response to Reviewer kMTD (Part 1).**
>
> ``` Waekness 1. The paper is not presented well and very empirical. I would suggest skip introducing too much details about preliminary works such as VQGAN, BAE without sharing insights designing them for image generation.```
>
> Thank you for your suggestion. We would like to clarify the main purpose and contributions of our work: This study is an **systematical empirical examination** aimed at exploring the applicability of standard language models in vision generation, bridging the gap between language and vision generation. Intentionally, we assume our work has a **broader audience**, hope it can inspired the exploration of language models in **other domains**. Therefore, we included a brief introduction to image tokenization methods like VQGAN and BAE to help readers unfamiliar with the vision domain understand how images can be tokenized. However, we appreciate your feedback and have made these sections more concise in the revised manuscript.
>
> ```Weakness 2. The main part of the paper is mainly about experimental comparisons but there are not comprehensive experiments made to support the claims.```
>
> Thank you for your feedback. An empirical examination aimed at exploring the applicability of standard language models in vision generation, we consciously **minimized domain-specific adjustments** to their architecture or methodology and resitrict the comparioson scope to provide a fair assessment of the abilities of token-prediction-based language models in the vision domain.
>
> We thoroughly explored the **critical components** such as tokenization choice, modeling choice, vocabulary design, and sampling strategy. Our evaluations were primarily conducted using the large-scale real-world dataset, ImageNet, to ensure robust testing. Additionally, to demonstrate the adaptability of our model, we included experiments on distinct datasets like the Descriptive Texture Dataset and CelebA in our revised manuscript.
>
> We believe our work highlights the versatility and robustness of traditional language modeling methods and hope it will inspire further research in developing unified multimodal content generation frameworks.
>
> ```Weakness 3. Only image quality is reflected in figures such as Figure 8, I would suggest also adding text prompts and also possibly give more complex cases to compare the language understanding ability of autoregressively based LM for visual generation.```
>
> Thank you for your suggestion. We evaluated our model's robustness in scenarios requiring conditional generation with complex conditions, such as class interpolation defined as $\alpha$A+(1−$\alpha$)B, A and B denotes two different class label, $\alpha \in [0,1]$. The results indicate that the model effectively **learns and adapts to** the conditional information, rather than simply memorizing it. For instance, when interpolating between similar classes like a golden retriever and a husky, the model produces images that blend features from both classes when $\alpha$ is around 0.5. Conversely, with dissimilar classes such as a horse and a beer bottle, the model predominantly generates images that reflect the features of the class with the greater weight. These findings and corresponding images are detailed in **Appendix A.1.1** of our revised manuscript.
>
> Regarding text prompts, we believe autoregressive models inherently possess strong language understanding capabilities and are currently training a text-to-image model. We believe that a thorough investigation of text-to-image generation, given its complexity and unique challenges, deserves a dedicated study, which we intend to pursue in future work.
>
> ```Weakness 4. The fonts are too small in figure 4,5,6,7.```
>
> Thanks for your suggestion! We have enlarged the fonts for all the figures in our revised manuscript. Please have a check if you like!
>
> ```Weakness 5. Only AR models are used as baselines in this paper.```
>
> Our primary focus in this work is to explore and enhance the performance of language models in the vision domain, so we primarily compared ELM with other methods that utilize language models. However, we appreciate your feedback and have added comparisons with popular and state-of-the-art (SOTA) diffusion models. These results are now included below, and **Table 2** in the revised manuscript has been updated to reflect this additional evaluation.
> | Model     | param. size | fid  | IS     | Precision | Recall |
> |-----------|-------------|------|--------|-----------|--------|
> | DiT-XL[1] | 675M        | 2.27 | 278.24 | 0.83      | 0.57   |
> | SiT-XL [2] | 675M        | 2.06 | 277.50 | 0.83      | 0.59   |
> | ELM-XL (ours)    | 757M        | 1.79 | 332.99 | 0.80      | 0.59   |
>
> [1]Scalable Diffusion Models with Transformers (ICCV2023)
>
> [2]SiT: Exploring Flow and Diffusion-based Generative Models with Scalable Interpolant Transformers (ECCV2024)

---

> > ### Author Response · Authors · 2024-11-19
> > **Official Response to Reviewer kMTD (Part 2).**
> >
> > ```Question 1. Overall, what do you think is the best strength of using LLM for visual generation than diffusion models?```
> >
> > The main advantage of using large language models (LLMs) for vision generation includes the **training-efficiency**, **scalability** and **adaptability**.
> >
> > - **training-efficiency**: We compare the w.o CFG FID score along with training steps for DiT and ELM, the results below showcase that ELM has higher training efficiency than DiT and can generate better results withe much lower FID compared to DiT a the early training stage.(~ denotes approximate results, as the original work only provided these data through line plots rather than precise values.)
> >
> > | Training step | 100k  | 200k  | 400k  | 800k  |
> > |---------------|-------|-------|-------|-------|
> > | DiT-L/2[1]    | ~51   | ~30   | 23.33 | ~20   |
> > | ELM-L         | 21.65 | 16.82 | 16.03 | 15.75 |
> > | DiT-XL/2[1]   | ~49   | ~25   | 19.47 | ~18   |
> > | ELM-XL        | 19.98 | 15.27 | 13.56 | 12.80 |
> >
> > - **scalability**: The scaling-law together with the higher training-efficiency, we believe language models hold higher potential in handling huge amount of training data with larger model size.
> >
> > - **adaptability**: Unlike diffusion models, which works in continuous space and specifically suitable for image processing, LMs serve as **a more versatile method by simply predicting the next token**, which allows them to be easily adapted across various modalities.
> >
> > This advantages highlight language models value for cross-modal AI applications, and we believe this work lays the groundwork for future advancements in unified multimodal content generation.
> >
> > ```Question 2: For the tokenizer selection, have you considered using the VAE?```
> >
> > We primarily focus on discrete tokenization methods in this work to **directly evaluate the adaptability of language models** in the vision domain by avoiding domain-specific adjustments. While we recognize that other studies, such as MAR[1], introduce specialized methods like combining diffusion loss with an autoregressive mechanism to enable continuous tokenization (e.g., VAE), our objective differs. We aim to provide a straightforward evaluation of language models' potential in vision tasks without introducing additional modifications, offering a complementary perspective to those more specialized designs.
> >
> > [1] Autoregressive Image Generation without Vector Quantization.

---

> > > ### Comment · Reviewer_kMTD · 2024-11-25
> > >
> > > Thanks for the reviewers for the detailed response. I think the novelty and contributions are still limited. Also the experiments are not comprehensive enough and general enough for support the claim. Therefore, I would like to keep my previous rating as prone to reject.

---

> ### Author Response · Authors · 2024-11-25
> **Follow-up**
>
> Dear Reviewer,
>
> Thank you again for the valuable comments. We have carefully addressed the main concerns you raised in detail. As the discussion phase is about to close, we look forward to any additional feedback you may have. We would be happy to clarify any further questions or concerns.
>
> Best,
> The authors

---

### Official Review · Reviewer_YWpg · 2024-10-29

**Soundness:** 3
**Presentation:** 2
**Contribution:** 2
**Rating:** 6
**Confidence:** 3

**Summary:**

This paper conducts a thorough analysis on the use of LMs for image generation. This approach usually involves two stages: (1) training an image quantizer to convert image patches into discrete tokens and (2) training a language model to model this token distribution. While this approach is not new, this paper dives deeper into the design choices used in this generation setup, providing insights into the architectural and hyperparameter choices that influence this regime. They demonstrate strong results on ImageNet generation.

**Strengths:**

- The paper explores many different dimensions of the LM setup, ablating across both the stage 1 tokenizer and the stage 2 LM.
- Section 3 of the paper provides nice visualizations and sheds some light on how LMs learn the image generation task (which is not entirely intuitive, especially for the autoregressive nature).
- This paper explores autoregressive generation, which has distinct advantages over diffusion models (for example, leveraging LLM infra and systems advances). Diffusion models have been the primary focus of the image generation community as of late, so further exploration into this paradigm is valuable.

**Weaknesses:**

- The evaluation results are only on 256px ImageNet images, which is saturated at this point. It would be more valuable if the results could be demonstrated on a different task, e.g., text-to-image generation on MS-COCO, or at least on other conditional generation datasets (e.g., CelebA or FFHQ). This would also ensure that the findings transfer to other datasets/tasks.
- The paper suggests that ELM can be used to generate any size images, but there doesn’t seem to be evaluations done at higher resolutions. How does the model compare to 512px images, for example? Is it significantly better to use ELM to generate at 512px, compared to resizing 256px generated images?
- All of the ablations are conducted on model architectures, but recent trends in deep learning suggest that data is more important than architecture. How do the LM models compare against diffusion models in terms of data efficiency/scale? This would be a valuable ablation to run, to find out if either model is more data efficient, or if either LM or diffusion models are better at small data regimes.

**Questions:**

- How does the inference speed of ELM compare to that of diffusion models?
- Will the models/code be open sourced?

---

> ### Author Response · Authors · 2024-11-19
> **Official Response to Reviewer YWpg (Part 1).**
>
> ```Weakness 1. The evaluation results are only on 256px ImageNet images, which is saturated at this point. It would be more valuable if the results could be demonstrated on a different task, e.g., text-to-image generation on MS-COCO, or at least on other conditional generation datasets (e.g., CelebA or FFHQ). This would also ensure that the findings transfer to other datasets/tasks.```
>
> Thank you for your feedback. Our initial evaluation focused systematically on 256px ImageNet images to **standardize(( comparisons and benchmark performance. However, acknowledging your suggestion, to address the **transferability** of our findings to other tasks and datasets, we conducted experiments on the CelebA dataset at 256px using the ELM-L with 2-8 tokenizer. The qualitative results (in **Appendix A1.3**) from these tests further substantiate the effectiveness of our model across different datasets.
>
> Regarding the text-to-image generation task, we believe autoregressive models inherently possess strong language understanding capabilities. However, a thorough investigation of text-to-image generation, given its complexity and unique challenges, deserves a dedicated study, which we intend to pursue in future work.
>
> ```Weakness 2. The paper suggests that ELM can be used to generate any size images, but there doesn’t seem to be evaluations done at higher resolutions. How does the model compare to 512px images, for example? Is it significantly better to use ELM to generate at 512px, compared to resizing 256px generated images?```
>
> Our claim that ELM can generate images of any size is based on its **inherent flexibility** as a next-token prediction model. This allows it to generate images with any sequence length without requiring adjustments to the model's architecture. For instance, even though the model is trained with a sequence length of 16*16 tokens on 256px images, it can seamlessly generate larger images by horizontally extending the canvas. In this way, the model maintains a receptive sequence length of 256 tokens but continues to predict additional tokens to expand the image size, effectively generating wider scenes.
>
> To specifically address your question about higher-resolution images, we have finetuned an ELM-L model on 512px ImageNet initialized with the model trained on 256px ImageNet. We found that the model adapts quickly and effectively to this higher resolution, achieving high-quality outputs within just 50 epochs. This rapid convergence and the quality of the generated 512px images demonstrate the model's robustness. The quantitative results are presented below. The qualitative results and discussion are shown in our revised paper (**Appendix A1.2**).
>
> | Model            | param. size | train. steps | fid  | IS     | Precision | Recall |
> |------------------|-------------|--------------|------|--------|-----------|--------|
> | DiT-XL/2[1]      | 675M        | 3000k        | 3.04 | 240.82 | 0.84      | 0.54   |
> | MaskGIT[2]       | 227M        | 1500k        | 7.32 | 156.0  | 0.78      | 0.50   |
> | ELM-L, 2-8(ours) | 312M        | 250k         | 4.82 | 246.87 | 0.81      | 0.59   |
>
>
> [1]Scalable Diffusion Models with Transformers (ICCV2023)
>
> [2]MaskGIT: Masked Generative Image Transformer (CVPR 2022)
>
> ```Weakness 3. All of the ablations are conducted on model architectures, but recent trends in deep learning suggest that data is more important than architecture. How do the LM models compare against diffusion models in terms of data efficiency/scale?...```
>
> Autoregressive language models (ARLMs) are widely recognized for their data efficiency, particularly in their ability to scale effectively with large amounts of training data, as supported by scaling laws established in prior studies. We have also conducted a preliminary evaluation to compare the training efficiency of diffusion models and ARLMs on ImageNet. The results, provided below, shows the **FID on 256*256 ImageNet w.o. CFG along with training steps**, which demonstrate that **ARLMs achieve higher training efficiency**, requiring fewer training iterations to produce high-quality results. (~ denotes approximate results, as the original work only provided data through line plots rather than precise values.)
>
> | Training step | 100k  | 200k  | 400k  | 800k  |
> |---------------|-------|-------|-------|-------|
> | DiT-L/2[1]    | ~51   | ~30   | 23.33 | ~20   |
> | ELM-L         | 21.65 | 16.82 | 16.03 | 15.75 |
> | DiT-XL/2[1]   | ~49   | ~25   | 19.47 | ~18   |
> | ELM-XL        | 19.98 | 15.27 | 13.56 | 12.80 |
>
> Furthermore, we tested the ELM on **smaller** datasets such as the Describable Textural Dataset, which contains approximately 5,000 images, and CelebA, which includes around 200,000 human face images. In both cases, the ELM achieved high-quality results within 400 epochs (see Appendix **A1.3** in the updated paper). This underscores the flexibility and robustness of AR models in both large and small data regimes.

---

> > ### Author Response · Authors · 2024-11-19
> > **Official Response to Reviewer YWpg (Part 2).**
> >
> > ```Question 1. How does the inference speed of ELM compare to that of diffusion models?```
> >
> > Regarding inference speed, diffusion models typically require around 50 sampling steps during inference (e.g., Stable Diffusion) to ensure high-quality generation. In contrast, autoregressive (AR) language models require a number of sampling steps equal to the sequence length (e.g., 16×16 steps for 256px images), which can result in longer inference times compared to diffusion models. However, masked language models (MLMs) can achieve competitive inference efficiency, requiring only about 10 sampling steps.
> >
> > Actually, inference efficiency is beyond the primary scope of this work, as our focus is on demonstrating the capabilities of language models in the vision domain. That said, numerous studies aim to accelerate AR model inference, such as the use of KV-cache and parallel decoding strategies, which are modality-agnostic and can be **readily applied to ARLMs in the vision domain**. Therefore, we believe that once the potential of ARLMs in other domains is established, many of these optimization techniques from the language domain can be adapted to enhance their practical deployment.
> >
> > ```Question 2. Will the models/code be open sourced?```
> >
> > All the models and code, including the additional experiments we conducted during the rebuttal period, will be released.

---

> > > ### Comment · Reviewer_YWpg · 2024-11-20
> > >
> > > Thank you for the detailed responses, and the new results! I think that including these results will substantially strengthen the paper. I've raised my score to reflect this.

---

> > > > ### Author Response · Authors · 2024-11-25
> > > > **Thanks for the reply**
> > > >
> > > > Thanks a lot for your reply! We are glad that we have solved your questions and concerns!

---

### Official Review · Reviewer_Pv6w · 2024-11-03

**Soundness:** 2
**Presentation:** 2
**Contribution:** 2
**Rating:** 3
**Confidence:** 4

**Summary:**

The paper investigates the design space of image generation using language models, whose space includes the type of image tokenizer (e.g., vector-quantized or binary-value quantized auto-encoders, decomposition), type of language model (e.g., autoregressive or masked language model), scaling behavior (e.g., learning vs model sizes, vocabulary size vs model size). With extensive study, the paper suggests an optimal combination of design choices, leading to a strong image generation performance on class-conditional image generation on 256x256 ImageNet, whose performance is on par with existing state-of-the-art methods.

**Strengths:**

* The paper has clearly laid out some important design choices of building language model based image generation model. The recipe provided in this paper could be useful to the community.

**Weaknesses:**

* Overall, I didn't find the findings of the paper new, surprising, or significant over those from previous works.

* The design space elucidated in this paper is mostly from existing works. While it is great that the paper has compiled them into a single paper, the contribution is very limited by nature. Furthermore, the optimal design choices made in this paper is not very different from the previous findings, which limit the contribution of the paper to confirm what is already known.

  * Section 3.2 Tokenizer choice: VQGAN vs BAE is repetition of the study conducted by previous work [Yu et al., 2024]
  * Section 3.5 Vocabulary design has been studied in [Yu et al., 2024] (Section 3.1, paragraph "Token factorization for efficient prediction") and confirmed decomposition into two subcodes being generally optimal.

* Section 3.1 Image generation vs text generation, which compares the property of image and text tokenizers, does not seem rigorous and conclusive.
  * In Table 1, how is "bigram" distribution obtained from the image tokens? Specifically, how is "consecutiveness" defined? Raster-order?
  * Token distribution analysis seems misleading. For example, the sentence in line 216 "the randomness in image token distribution implies that image generation doesn't depend on strict sequential patterns" is not well justified. Unigram distribution being uniform only suggests that the frequency of items in vocabulary is uniformly distributed, but it is unclear how unigram distribution could imply anything about the sequential pattern.
  * Ultimately, what is the take-home message of this section? Image and text tokenizers have different distribution and image tokenizer seems to have more randomness, so it is more difficult. And? What design choice consideration should we be making knowing that the difference in distributions?

[Yu et al., 2024] [LANGUAGE MODEL BEATS DIFFUSION — TOKENIZER IS KEY TO VISUAL GENERATION](https://arxiv.org/pdf/2310.05737)

**Questions:**

Please see weakness section for more context.

* What are new findings from Section 3.2 and 3.5 compared to [[Yu et al., 2024](https://arxiv.org/pdf/2310.05737)]?
* Regarding Section 3.1:
  * In Table 1, how is "bigram" distribution obtained from the image tokens? Specifically, how is "consecutiveness" defined? Raster-order?
  * The sentence in line 216 "the randomness in image token distribution implies that image generation doesn't depend on strict sequential patterns" is not well justified. Unigram distribution being uniform only suggests that the frequency of items in vocabulary is uniformly distributed, but it is unclear how unigram distribution could imply anything about the sequential pattern.
  * Ultimately, what is the take-home message of the section 3.1? Image and text tokenizers have different distribution and image tokenizer seems to have more randomness, so it is more difficult. And? What design choice consideration should we be making knowing that the difference in distributions?

---

> ### Author Response · Authors · 2024-11-19
> **Official Response to Reviewer Pv6w (Part 1).**
>
> ```Weakness 1. Overall, I didn't find the findings of the paper new, surprising, or significant over those from previous works.```
>
> Thanks for your feedback. We would like to clarify the primary objectives and contributions of our work. We aim to present a **comprehensive empirical study exploring the potential of standard language models in vision generation**, intentionally minimizing adjustments to their architecture or methodology to fairly assess the ability of token-prediction-based language models in vision domain.
>
> While some findings may align with existing insights, the significance of our study lies in its **systematic exploration** of language models' capabilities in vision tasks. Our work methodically **identifies optimal strategies** and **directly assesses the versatility and robustness** of traditional language modeling methods.
>
> Our work sets a strong methodological foundation for **bridging the gap between language and vision generation**, aims to inspire further research in developing unified multimodal content generation frameworks.
>
> Below, we would like to address your specific concerns to illustrate our contribution.
>
> ```Weakness 2. Section 3.2 Tokenizer choice: VQGAN vs BAE is repetition of the study conducted by previous work [Yu et al., 2024], Section 3.5 Vocabulary design has been studied in [Yu et al., 2024]```
>
> Thank you for your observations and critique. We would like to reiterate that the primary goal of our work is to serve as a valuable resource by compiling and synthesizing the exploration of language models in the vision domain into a unified framework. Our detailed experiments and systematic exploration provide novel insights and actionable strategies that advance the understanding of design choices in this field. Below, we address your specific concerns in detail:
>
> - **Tokenizer choice (Section 3.2)**: While [Yu et al., 2024] introduced a look-up-free quantization method (a deterministic version of BAE from [Wang et al., 2023]) and compared it with VQGAN, their focus was largely on redesigning the **tokenizer's model architecture** to enhance performance on video generation. In contrast, we utilize the standard 2D-convolution-based model architecture of VQGAN, as described in [Esser et al., 2021], to ensure **a fair and controlled comparison between BAE and VQGAN**. Additionally, we extend the exploration by **incorporating Bernoulli sampling** strategies within the BAE framework and analyzing **the impact of this component on image generation**. These distinctions underline that our evaluation of tokenizer choices is an independent and nuanced study rather than a replication of [Yu et al., 2024].
> - **Vocabulary design (Section 3.5)**: While [Yu et al., 2024] suggested token factorization into two subcodes, their experiments primarily focused on an 18-dimensional code **without** a thorough investigation of factorization performance or the broader implications of varying configurations. Moreover, they did **not** explicitly determine which design strategy is optimal. In our work, we systematically explore the impact of subcode configurations, including **varying the number of subcodes (beyond two) and their dimensionality**. Through extensive empirical studies, we provide clear conclusions on the optimal strategy for vocabulary design. Furthermore, we employ **a different subtoken aggregation method** by concatenating subtokens followed by projection, which extends the simple summation approach used in [Yu et al., 2024].
>
> We deeply admire the contributions of [Yu et al., 2024] and appreciate the inspiration their work provides. However, we believe our study distinguishes itself by offering **a broader and more detailed examination of design choices**, which enriches the understanding of token-prediction-based language models in image generation.
>
> [Yu et al., 2024] Language model beats diffusion - tokenizer is key to visual generation. ICLR 2024.
>
> [Wang et al., 2023] "Binary latent diffusion." CVPR 2023.
>
> [Esser et al., 2021] "Taming transformers for high-resolution image synthesis." CVPR 2021.

---

> ### Author Response · Authors · 2024-11-19
> **Official Response to Reviewer Pv6w (Part 2).**
>
> ```Weakness 3. (1) In Table 1, how is "bigram" distribution obtained from the image tokens? Specifically, how is "consecutiveness" defined? Raster-order? (2) Token distribution analysis seems misleading. For example, the sentence in line 216 "the randomness in image token distribution implies that image generation doesn't depend on strict sequential patterns" is not well justified. Unigram distribution being uniform only suggests that the frequency of items in vocabulary is uniformly distributed, but it is unclear how unigram distribution could imply anything about the sequential pattern. (3) Ultimately, what is the take-home message of this section? ```
>
> Thank you for your thoughtful comments. The comparison of image and text tokens in Section 3.1 is designed to offer an **intuitive explanation** of the unique challenges and phenomena associated with adapting language models to the vision domain. While we acknowledge that the explanation might not possess the formal rigor typically expected, our aim was to provide an **accessible understanding of the robustness** of token-prediction-based language models across distinct modalities. Below, we address your specific concerns in detail.
>
> - **Bigram distribution and "consecutiveness"**: The "consecutiveness" in Table 1 is derived from raster-ordered image tokens, where the tokens are arranged sequentially row by row. We intentionally chose raster ordering to minimize domain-specific design choices, as our goal was to evaluate the robustness of token-prediction-based language models under general conditions.
>
> - **Token distribution analysis and its implications**: The KL divergence between the uniform distribution and the unigram token distribution underscores the inherent randomness in the token distribution of large-scale, real-world image datasets. This randomness highlights the lack of structured patterns typically seen in text tokenization. To complement this, **the bigram distribution is analyzed to partially reflect sequential patterns**. In the updated revision, we further include the **perplexity of n-gram models**, which further indicates a lack of orderliness in the vision domain compared to the language domain.
>
> - **Take-home message**: As the initial phase of our exploration into the design space, **this section highlights the robustness of language models in the image domain**. Despite the distinct input structures and training dynamics inherent to this domain, language models effectively learn features that enable the generation of high-quality images. This adaptability underscores their robustness and versatility, making them promising candidates for multi-modal content generation frameworks. Moreover, these findings **establish the foundation and necessity for the further exploration** of other critical aspects within the design space.

---

> ### Author Response · Authors · 2024-11-25
> **Follow-up**
>
> Dear Reviewer,
>
> Thank you again for the valuable comments. We have carefully addressed the main concerns you raised in detail. As the discussion phase is about to close, we look forward to any additional feedback you may have. We would be happy to clarify any further questions or concerns.
>
> Best,
> The authors

---

> > ### Comment · Reviewer_Pv6w · 2024-11-27
> >
> > Thank authors for their response in confirming the contribution of the work and additional experiments. I still do not think the token distribution analysis provides constructive insight or direction as it only confirms the effectiveness of language model in vision token modeling, which we've seen many successful examples. I might be missing, but I think the paper does not provide insight on why or how language models manage to learn from such random patterns or ways to design image tokenizer with more favorable distributions to language models.

---

### Official Review · Reviewer_WfML · 2024-11-03

**Soundness:** 2
**Presentation:** 2
**Contribution:** 2
**Rating:** 5
**Confidence:** 3

**Summary:**

In this work, the authors investigate the effects of different modeling choices for using language models for image generation. Specifically, they compare two tokenizers (VQGAN and BAE), two loss functions (AR and MLM), vocabulary designs, and different sampling strategies. Based on this exploration, the authors propose the ELM model and achieve strong performance on the class-conditional ImageNet 256X256 image generation task when compared with competitive AR baselines.

**Strengths:**

1.	Research questions proposed by this paper, i.e. how to design language models for image generation, is a very timely and important subject to investigate.

2.	This paper presents substantial efforts in searching for better alternatives compared to the dominating tokenization approach that uses a pretrained VQGAN model.

3.	Their proposed ELM achieves SOTA performance on class-conditional ImageNet 256X256 image generation.

**Weaknesses:**

While this paper explores an important research direction, the attempt is nevertheless lacks rigor and stronger experimental validation. Specifically, my main concerns include the following:

1.	Failure to compare with continuous-valued tokenization: As the most competitive baseline shown in this paper, MAR, which uses continuous-valued tokenization, achieves very comparable performance (FID 1.55) in comparison to ELM (FID 1.54). However, the authors did not include continuous-valued tokenizers in their design space.

2.	Failure to compare with masked autoregressive loss: Similarly, MAR uses masked autoregressive loss, which is also not included in their design space.

3.	Unconvincing comparison with MAR: Note that MAR  achieves almost identical performance as the best performing ELM with only half of the parameter size. And when comparing the two models with similar parameter size, MAR shows very clear advantages over ELM. This shows the minor improvement that ELM brings – especially with the discussion of scaling laws, one can imagine scaling up MAR further can potentially also improve the performance, and therefore MAR can potentially outperform ELM if scaled up to 2B parameter size.

4.	Failure to explore the effect of token orderings and token types: Even if the authors intend to only compare discrete tokens, they have missed several comparison points such as the ordering of the tokens (e.g. scanline, zig-zag, etc) and the types of the tokens (e.g. image patches, image scales, etc). These design choices are arguably equally important and underexplored as the ones presented in the paper.

5.	Contradicting discussions and conclusions: In Section 3.1, the authors noted that training loss is not a good indicator for the model performance. However, in later sections, e.g. when analyzing the scaling laws, the authors still use the training loss as the metric for evaluation.

6.	A few claims:
a.	In Section 3.1, the authors claim that “image data lacks the inherent structure” from their KL divergence analysis with uniform distribution. Not only does this analysis have no connection to the conclusion, the claim itself is also invalid without further assumptions (e.g. [1]). If the authors had explored different token types (e.g. the scale tokens), they may also observe different divergences and may reach different conclusions.
b.	In Section 3.4, the authors use the visualized attention maps to study the ability to learn global v.s. local information. However, this kind of visualization does not necessarily accurately reflect the actual functionality of the network according to [2].

7.	Failure to explore the combinatorial design space: The authors fix the tokenizer choice when comparing different loss functions, tokenization designs and sampling strategies. However, they fail to consider the combinations of these choices with the other tokenizer choice. For example, it is possible that VQGAN tokens + MLM behaves differently with AR and MLM, or even potentially performs better than BAE tokens + AR. Given that VQGAN is a popular choice for many AR image generation models, it would be valuable to analyze behaviors with VQGAN tokens.

A few minor suggestions:
1.	Line 164, the notation of the conditional probability is very confusing and not conventional
2.	Almost all plots have fonts that are too small and therefore hard to read

Overall, as a paper that claims to “elucidate the design space of language modeling for image generation”, the explored design space in this paper is not very comprehensive and the authors miss multiple obvious marks that have high likelihood of further improving the performance.

References:
[1] M. A. Turk and A. P. Pentland, "Face recognition using eigenfaces," CVPR 1991.
[2] Wen, Kaiyue, et al. "Transformers are uninterpretable with myopic methods: a case study with bounded Dyck grammars." NeurIPS 2024.

**Questions:**

It would be great if the authors can address the weakness mentioned above, especially a more thorough discussion and comparison with MAR.

---

> ### Author Response · Authors · 2024-11-19
> **Official Response to Reviewer WfML (Part 1).**
>
> ```Weakness 1&2&3, regarding to the comparison with MAR, including 1. continuous-valued tokenization, 2. masked autoregressive loss and 3. unconvincing comparison with MAR```
>
> For weakness 1 &2 &3, our primary goal with this research was to explore the potential of language models in the vision domain, aiming to pave the way for a unified framework for multi-modal content generation. In pursuit of **consistency with traditional language model methodologies**, we opted for **discrete tokenization** methods and a **classification-based training objective**. This choice aligns with **our intention to directly leverage the established practices within language modeling to the vision domain**, rather than adopting continuous tokenization or masked autoregressive loss, which are more characteristic of MAR.
>
> As discussed in Appendix (A.5 in our original paper, A6. in our updated paper) under 'Limitations’, while we recognize the advantages of methods like MAR in tailoring autoregressive models for image generation, our approach tries to provide a **straightforward evaluation** of **standard language models'** potential in vision tasks without introducing additional modifications. Regarding the model size comparison, although our model's largest capacity at 2 billion parameters exceeds that of MAR's; This discrepancy was not intended to disadvantage MAR but to **explore the scalability and potential of our method**.
>
> ```Weakness 4. Failure to explore the effect of token orderings and token types: Even if the authors intend to only compare discrete tokens, they have missed several comparison points such as the ordering of the tokens (e.g. scanline, zig-zag, etc) and the types of the tokens (e.g. image patches, image scales, etc). These design choices are arguably equally important and underexplored as the ones presented in the paper.```
>
> Thank you for your insightful comments regarding the exploration of token orderings and types. In response, we have conducted additional experiments to examine different token orderings, including a zig-zag pattern, using our L-sized autoregressive model equipped with a 2-8 BAE tokenizer and VQGAN tokenizer with 16384 codes. We compared these results with those from models using a scanline ordering, and the quantitative outcomes are detailed below.
>
> | Token order | tokenizer   | fid  | IS     | Precision | Recall |
> |-------------|-------------|------|--------|-----------|--------|
> | scan-line   | BAE 2-8    | 2.34 | 281.29 | 0.82      | 0.56   |
> | zig-zag     | BAE 2-8   | 2.69 | 257.73 | 0.82      | 0.56   |
> | scan-line   | VQGAN-16384 | 6.21 | 234.51 | 0.79      | 0.49   |
> | zig-zag     | VQGAN-16384 | 6.91 | 237.87 | 0.80      | 0.49   |
>
> Regarding the types of tokens, our approach primarily explores the method akin to the handling of text in language models, that is, treating the entire image as a 'sentence' where patches represent 'words' that compose the image. **This analogy aligns with our goal to parallel the processing of images to that of text in traditional language models**. While we recognize and value the specialized tokenization strategies explored in methods like VAR, as discussed in the Appendix under 'Limitation,' we have chosen to restrict our exploration to the design space most analogous to textual tokenization. We believe this focus is crucial for the scope of this work, which aims to **integrate language model methodologies into image generation more seamlessly**.
>
> ```Weakness 5. Contradicting discussions and conclusions: In Section 3.1, the authors noted that training loss is not a good indicator for the model performance. However, in later sections, e.g. when analyzing the scaling laws, the authors still use the training loss as the metric for evaluation.```
>
> Thank you for your observation regarding the use of training loss in our analysis. I’d like to clarify that these discussions address different aspects of training dynamics and are not contradictory, but serve different purposes.
>
> Firstly, in Section 3.1, we discuss the **implications of training loss on generation quality**. We highlight a phenomenon specific to training token-prediction mechanisms in the vision domain: even during the later training stages, a relatively high training loss can coexist with visually sound generated results. This observation arises from the **inherent randomness** in the patchified token distribution of large-scale real-world image datasets, **making a high training loss a reasonable outcome in such scenarios**.
>
> Secondly, when analyzing scaling laws, we use **training loss as an indicator of a model's capacity to fit the data**. Larger models demonstrate a stronger ability to fit the training data and therefore exhibit lower training losses compared to smaller models. However, due to **the dataset's intrinsic complexity** and **the randomness of the token distribution**, **the training loss remains high overall, regardless of model size**.

---

> ### Author Response · Authors · 2024-11-19
> **Official Response to Reviewer WfML (Part2).**
>
> ``` Weakness 6.1 a. In Section 3.1, the authors claim that “image data lacks the inherent structure” from their KL divergence analysis with uniform distribution. Not only does this analysis have no connection to the conclusion, the claim itself is also invalid without further assumptions (e.g. [1]). If the authors had explored different token types (e.g. the scale tokens), they may also observe different divergences and may reach different conclusions. ```
>
> We would like to clarify our intention and address your concerns.
>
> Firstly, our analysis of the KL divergence between the token distribution and a uniform distribution, as well as the bigram distribution, was aimed at providing an **intuitive explanation** for why a relatively high training loss can coexist with visually sound generated results. **The KL divergence measures the randomness of the token distribution**, while the **bigram analysis partially reflects token continuity**. Together, these offer insights into why the training loss remains high, particularly in the context of **large-scale real-world** datasets with **patch-based** tokenization. We also add the perplexity of n-gram models in **Table 1** in the revised manuscript, which further shows the lack of orderliness in the vision domain compared to language domain.
>
> We pointed out in the paper that this analysis is based on ImageNet— **a large-scale real-world** dataset. We acknowledge that our claim in Section 3.1 was not formally established with strict assumptions and that it may not generalize to all datasets or tokenization methods. For instance, as you pointed out, datasets like face recognition in [1] or alternative tokenization approaches, such as scale-based tokenization in VAR, could exhibit very different token distributions and corresponding divergences.
>
> In light of your suggestion, we will further clarify in the revised manuscript that our analysis and conclusions are based specifically on **large-scale real-world datasets**, such as ImageNet, using **patch-based tokenization**.
>
> ```Weakness 6.2 In Section 3.4, the authors use the visualized attention maps to study the ability to learn global v.s. local information. However, this kind of visualization does not necessarily accurately reflect the actual functionality of the network according to [2].```
>
> Thank you for your insightful comment and for referencing [2]. While [2] provides a theoretical argument that transformers are not inherently interpretable, it is important to note that their conclusion is derived under a **synthetic setup**, learning an **idealized formal language**, i.e., Dyck grammars, as a case study. As such, their findings, while valuable, may have limitations when applied to real-world tasks or more complex input domains.
>
> At the same time, **recent research, particularly in the context of large language models, has demonstrated the utility of attention maps in interpreting model behavior**. These studies have not only provided insights into model functionality but have also inspired effective strategies to enhance performance based on attention-based interpretations, like [3][4][5][6].
>
> In the vision domain, we believe that attention map visualization can **similarly offer valuable and straightforward insights into model behavior**. While we recognize the limitations of such visualizations as definitive proof of a network's functionality, they remain a useful tool for exploring how models process local versus global information, providing a basis for further analysis and refinement.
>
> [1] M. A. Turk and A. P. Pentland, "Face recognition using eigenfaces," CVPR 1991.
>
> [2] Wen, Kaiyue, et al. "Transformers are uninterpretable with myopic methods: a case study with bounded Dyck grammars." NeurIPS 2024.
>
> [3]Xiao G, Tian Y, Chen B, et al. "Efficient streaming language models with attention sinks." ICLR 2024.
>
> [4] Yu Z, Wang Z, Fu Y, et al. “Unveiling and harnessing hidden attention sinks: Enhancing large language models without training through attention calibration”. arXiv 2024.
>
> [5] Gu X, Pang T, Du C, et al. “When Attention Sink Emerges in Language Models: An Empirical View”. arXiv 2024.
>
> [6] Sun M, Chen X, Kolter J Z, et al. “Massive activations in large language models.” COLM 2024

---

> ### Author Response · Authors · 2024-11-19
> **Official Response to Reviewer WfML (Part3).**
>
> ```Weakness 7. Failure to explore the combinatorial design space: The authors fix the tokenizer choice when comparing different loss functions, tokenization designs and sampling strategies. However, they fail to consider the combinations of these choices with the other tokenizer choice. For example, it is possible that VQGAN tokens + MLM behaves differently with AR and MLM, or even potentially performs better than BAE tokens + AR. Given that VQGAN is a popular choice for many AR image generation models, it would be valuable to analyze behaviors with VQGAN tokens.```
>
> Given the exponential growth in the number of experiments required to test all possible combinations, we opted for a more practical approach: comparing the design choices for each step independently and then combining the optimal choices to evaluate their overall performance.
>
> Acknowledging the popularity of VQGAN in token-based image generation models, we have conducted additional experiments to assess the performance of VQGAN tokens with masked language modeling (MLM). **The result shown below aligns with the findings in our paper**, i.e.,  BAE combined with autoregressive (AR) modeling works best.
>
> | Modeling | tokenizer   | fid  | IS     | Precision | Recall |
> |----------|-------------|------|--------|-----------|--------|
> | AR       | BAE 1-16    | 2.38 | 271.54 | 0.82      | 0.56   |
> | AR       | VQGAN-16384 | 6.21 | 234.51 | 0.79      | 0.49   |
> | MLM      | BAE 1-16    | 3.67 | 272.23 | 0.85      | 0.47   |
> | MLM      | VQGAN-16384 | 7.54 | 235.56 | 0.80      | 0.49   |
>
> ```Suggestion. 1. Line 164, the notation of the conditional probability is very confusing and not conventional 2. Almost all plots have fonts that are too small and therefore hard to read.```
>
> Thanks for your suggestion! We have revised the notation in line 164 (line 160 in the revised manuscript) and enlarged the fonts in the plots! Please have a look! Thanks a lot!

---

> > ### Comment · Reviewer_WfML · 2024-11-25
> >
> > I thank the authors for providing a detailed feedback. However, given the amount of additional changes that the authors need to make, I think that the paper would benefit from an additional round of revision and an additional round of reviews.

---

> ### Author Response · Authors · 2024-11-25
> **Follow-up**
>
> Dear Reviewer,
>
> Thank you again for the valuable comments. We have carefully addressed the main concerns you raised in detail. As the discussion phase is about to close, we look forward to any additional feedback you may have. We would be happy to clarify any further questions or concerns.
>
> Best,
> The authors

---

### Official Review · Reviewer_3q1x · 2024-11-04

**Soundness:** 2
**Presentation:** 3
**Contribution:** 2
**Rating:** 5
**Confidence:** 3

**Summary:**

This paper investigates the application of language models for vision generation tasks by exploring differences between image and text token distributions, the training dynamics of autoregressive models versus masked language models (MLMs), and the efficacy of different image discretization approaches. The authors propose a new model, ELM (Elucidated Language Model for Image generation), which combines AR modeling with Binary Autoencoder (BAE) for discretization. Key insights include the advantage of AR models for capturing image structures, the benefits of BAE in reducing computational costs and improving performance, and the ability of AR models to learn effective image patterns without inductive biases.

**Strengths:**

The paper examines different components of language models in image generation, such as tokenization methods, model scalability, and sampling strategies, providing insights into optimizing language models for visual tasks.

The ELM model integrates binary autoencoders for effective tokenization and AR models for scalability and performance. Extensive experimentation validates the authors' design choices, as ELM achieves high performance across various model sizes.

**Weaknesses:**

While the paper positions ELM as an alternative to diffusion models for image generation, more direct comparisons with these models could strengthen the evaluation. How does ELM compare in performance and efficiency to diffusion models in a similar setting?

While the study uses ImageNet as a benchmark, it doesn’t explore a broader range of datasets or domains that might reveal limitations in model generalization. How might the model’s performance be affected by training on datasets with higher resolution images or more complex scenes than ImageNet provides.

How robust is ELM in scenarios requiring conditional generation with complex prompts or context?

**Questions:**

Could the proposed design principles for image generation apply to other visual tasks, such as video generation?

How does the model handle diverse visual patterns, such as textures or irregular structures, compared to more common AR or diffusion models?

---

> ### Author Response · Authors · 2024-11-19
> **Official Response to Reviewer 3q1x (Part 1).**
>
> Thanks for your valuable suggestion! Below, we answer your concern and questions in detail.
>
> ```Weakness 1. While the paper positions ELM as an alternative to diffusion models for image generation, more direct comparisons with these models could strengthen the evaluation. How does ELM compare in performance and efficiency to diffusion models in a similar setting?```
>
> Our primary focus in this work is to explore and enhance the performance of language models in the vision domain, so we primarily compared ELM with other methods that utilize language models. However, we appreciate your feedback and have added comparisons with popular and state-of-the-art (SOTA) diffusion models. These results are now included below, and **Table 2** in the revised manuscript has been updated to reflect this additional evaluation.
> | Model     | param. size | fid  | IS     | Precision | Recall |
> |-----------|-------------|------|--------|-----------|--------|
> | DiT-XL[1] | 675M        | 2.27 | 278.24 | 0.83      | 0.57   |
> | SiT-XL [2] | 675M        | 2.06 | 277.50 | 0.83      | 0.59   |
> | ELM-XL (ours)    | 757M        | 1.79 | 332.99 | 0.80      | 0.59   |
>
>
> As for **effieciency**, we compare the **training efficiency** of ELM with DiT under similar training set. The results, provided below, shows the **FID on 256*256 ImageNet w.o. CFG along with training steps**, which demonstrate that autoregressive language models achieve higher efficiency, requiring fewer training iterations to produce high-quality results. (~ denotes approximate results, as the original work only provided data through line plots rather than precise values.)
> | Training step | 100k  | 200k  | 400k  | 800k  |
> |---------------|-------|-------|-------|-------|
> | DiT-L/2[1]    | ~51   | ~30   | 23.33 | ~20   |
> | ELM-L         | 21.65 | 16.82 | 16.03 | 15.75 |
> | DiT-XL/2[1]   | ~49   | ~25   | 19.47 | ~18   |
> | ELM-XL        | 19.98 | 15.27 | 13.56 | 12.80 |
>
> ```Weakness 2. While the study uses ImageNet as a benchmark, it doesn’t explore a broader range of datasets or domains that might reveal limitations in model generalization. How might the model’s performance be affected by training on datasets with higher resolution images or more complex scenes than ImageNet provides.```
>
> While ImageNet is a widely used benchmark that includes a diverse range of categories, we agree that evaluating the model on a broader set of datasets could provide deeper insights into its generalization capabilities. In response, we have also tested our model on CelebA, a human face dataset, and Describable Textures Dataset (DTD) that evolving collection of textural images in the wild (also for your Question2), to assess its performance across a distinct domain. Our model can effectively train on these datasets, and results are added in our revised manuscript in **Appendix A1.3**.
>
> Regarding higher-resolution datasets, the primary focus of this work is to explore the optimal design strategy for language models in image generation and their potential in various domains. To efficiently investigate different design choices, we chose to work with lower-resolution images. As for higher-resolution datasets, we trained an ELM-L model on 512$\times$512 ImageNet for only 50 epochs (250k steps) initialized with the model trained on 256$\times$256 ImageNet, the quantitative results are presented below. The qualitative results and discussion are shown in our revised paper (**Appendix A1.2**).
> | Model            | param. size | train. steps | fid  | IS     | Precision | Recall |
> |------------------|-------------|--------------|------|--------|-----------|--------|
> | DiT-XL/2[1]      | 675M        | 3000k        | 3.04 | 240.82 | 0.84      | 0.54   |
> | MaskGIT[3]       | 227M        | 1500k        | 7.32 | 156.0  | 0.78      | 0.50   |
> | ELM-L, 2-8(ours) | 312M        | 250k         | 4.82 | 246.87 | 0.81      | 0.59   |
>
> [1]Scalable Diffusion Models with Transformers (ICCV2023)
>
> [2]SiT: Exploring Flow and Diffusion-based Generative Models with Scalable Interpolant Transformers (ECCV2024)
>
> [3]MaskGIT: Masked Generative Image Transformer (CVPR 2022)

---

> ### Author Response · Authors · 2024-11-19
> **Official Response to Reviewer 3q1x (Part 2).**
>
> ```Weakness 3. How robust is ELM in scenarios requiring conditional generation with complex prompts or context?```
>
> We evaluated our model's robustness in scenarios requiring conditional generation with complex prompts, such as class interpolation defined as $\alpha$A+(1−$\alpha$)B, A and B denotes two different class label, $\alpha \in [0,1]$. The results indicate that the model effectively learns and adapts to the conditional information, rather than simply memorizing it. For instance, when interpolating between similar classes like a golden retriever and a husky, the model produces images that blend features from both classes when $\alpha$ is around 0.5. Conversely, with dissimilar classes such as a horse and a beer bottle, the model predominantly generates images that reflect the features of the class with the greater weight. These findings and corresponding images are detailed in Appendix A.1.1 of our revised manuscript.
>
> As for complex prompts, we believe autoregressive models inherently possess strong language understanding capabilities. However, a thorough investigation of text-to-image generation, given its complexity and unique challenges, deserves a dedicated study, which we intend to pursue in future work.
>
> ```Question 1. Could the proposed design principles for image generation apply to other visual tasks, such as video generation?```
> The primary objective of this study is to explore the potential of language models within the vision domain, with image generation serving as an initial step. By identifying effective strategies for token-prediction language models in image generation, we believe these approaches can be readily adapted to video generation. Conceptually, video frames can be treated as sequential images; thus, generating videos could be approached by concatenating these frames into a super long sequence—a method well-established in language processing. Given that our results affirm the robust capabilities of language models in image generation, we are confident that their attributes, such as generating content of indefinite length[4], can also be harnessed for video generation.
>
> ```Question 2. How does the model handle diverse visual patterns, such as textures or irregular structures, compared to more common AR or diffusion models?```
>
> To address this, we utilized the Descriptive Texture Dataset (DTD), which features 74 distinct texture types, to train an ELM-L with a 2-8 tokenizer. Our results, as presented in **Appendix A1.3** of our revised paper, demonstrate that **our model effectively handles diverse visual patterns, including textures and irregular structures**.
>
> [4] Efficient Streaming Language Models with Attention Sinks. (ICLR 2024)

---

> ### Author Response · Authors · 2024-11-25
> **Follow-up**
>
> Dear Reviewer,
>
> Thank you again for the valuable comments. We have carefully addressed the main concerns you raised in detail. As the discussion phase is about to close, we look forward to any additional feedback you may have. We would be happy to clarify any further questions or concerns.
>
> Best,
> The authors

---

### Meta-Review · Area_Chair_QPgx · 2024-12-25

**Metareview:**

The paper explores the design space of language models for image generation, systematically investigating key aspects such as tokenizer choice, model scalability, sampling strategies, and vocabulary design. It then proposes Elucidated Language Model (ELM) as a potential alternative to diffusion models. However, there are common concerns regarding limited novelty and the incomplete exploration of the design space Considering the substantial additional changes required to address these issues, I believe the paper would benefit from another round of revision. Therefore, I recommend rejection at this stage.

**Additional Comments On Reviewer Discussion:**

The discussion reveals several critical points that the current submission misses, including novelty, the lack of ablation on important design choices, a lack of comprehensive comparisons with diffusion models, limited generality across datasets, and insufficient exploration of resolution and scalability. While the authors partially addressed these concerns during the rebuttal phase, addressing them fully would require substantial changes to the original paper and a large number of additional results that were not initially provided.

---

### Decision · Program_Chairs · 2025-01-22

Reject